# Inherent regulatory asymmetry emanating from network architecture in a prevalent autoregulatory motif

Md Zulfikar Ali[1,2†], Vinuselvi Parisutham[1,2†], Sandeep Choubey[3], Robert C Brewster[1,2]*

[1]Program in Systems Biology, University of Massachusetts Medical School, Worcester, United States; [2]Department of Microbiology and Physiological Systems, University of Massachusetts Medical School, Worcester, United States; [3]Max Planck Institute for the Physics of Complex Systems, Dresden, Germany

**Abstract** Predicting gene expression from DNA sequence remains a major goal in the field of gene regulation. A challenge to this goal is the connectivity of the network, whose role in altering gene expression remains unclear. Here, we study a common autoregulatory network motif, the negative single-input module, to explore the regulatory properties inherited from the motif. Using stochastic simulations and a synthetic biology approach in *E. coli*, we find that the TF gene and its target genes have inherent asymmetry in regulation, even when their promoters are identical; the TF gene being more repressed than its targets. The magnitude of asymmetry depends on network features such as network size and TF-binding affinities. Intriguingly, asymmetry disappears when the growth rate is too fast or too slow and is most significant for typical growth conditions. These results highlight the importance of accounting for network architecture in quantitative models of gene expression.

*For correspondence:
Robert.brewster@umassmed.edu

†These authors contributed equally to this work

Competing interests: The authors declare that no competing interests exist.

## Introduction

The genomics revolution has enabled biology with the ability to read, write and assemble DNA at the genome scale with single base pair resolution. These advancements have provided an important tool for the field of gene regulation that aims to predict gene expression from the regulatory code, inscribed in DNA (*Carey et al., 2013*; *Kosuri et al., 2013*; *Sharon et al., 2012*). This approach relies on quantitative measurements of gene expression as the regulatory DNA is systematically designed to induce regulation by various transcription factors (TFs) at specific positions or with differing affinities. However, success in predicting expression levels of natural genes from sequence alone has been relatively modest. One obvious complication is that genes are not isolated but rather exist in dense, interconnected networks. The concept of network motifs, defined as overrepresented patterns of connections between genes and TFs in the network, helps to digest these large networks into smaller subgraphs with specific properties; each of these motifs can be interpreted as performing a particular 'information processing' function that is determined by the connectivity and regulatory role of the genes in the motif (*Alon, 2006*; *Alon, 2007*; *Davidson, 2006*; *Mangan and Alon, 2003*; *Tkacik et al., 2008*). In this study, we dissect a prevalent gene regulation motif, the single-input module (SIM), to demonstrate the influence of network size and connectivity on the regulation of a network motif.

The SIM is a network motif where a single TF regulates the expression of a set of genes, including itself (*Figure 1A*). In *E. coli*, this motif is prevalent; the majority of TFs are autoregulated and have multiple targets (*Santos-Zavaleta et al., 2019*). Typically, this group of genes have related functions and the purpose of this motif is to coordinate, in both time and magnitude, expression of these

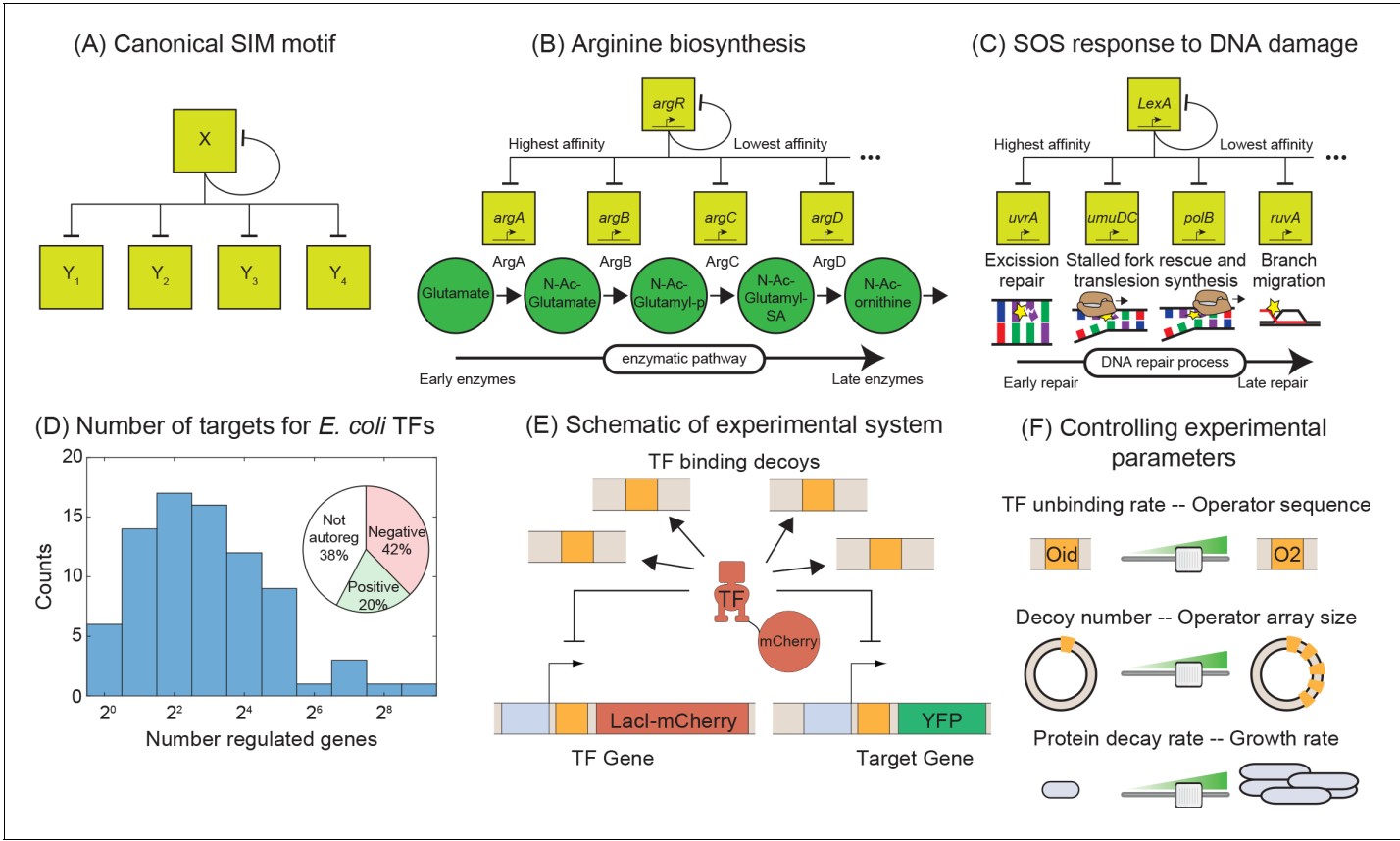

**Figure 1.** Synthetic approach to exploring the negative SIM motif. (**A**) Schematic of a canonical SIM motif: A single TF regulates itself and several other genes. (**B and C**) Examples of SIM motifs in *E. coli*. (**B**) ArgR is a transcriptional regulator of arginine biosynthesis. It auto-regulates itself and genes involved in different steps of arginine biosynthesis with precision in expression starting from the first enzyme of the pathway down to the last. This precise ordering is thought to originate from a corresponding ordering in TF-binding affinities of the target genes. (**C**) LexA is the master regulator of SOS pathway and is actively degraded in response to DNA damage. LexA auto-represses itself and represses a set of other genes involved in DNA repair. In this case, the early response genes have low affinity for the repressor while the late acting genes have high affinity, enabling temporal ordering of the response. (**D**) Histogram showing the number of known regulated genes for every TF in *E. coli*. Inset shows different modes of regulation of the TF genes. 62% of the TF genes are autoregulated with 42% negatively autoregulated and 20% positively auotregulated. (**E**) Schematic of the experimental model of a SIM motif used in this study. Here, LacI-mCherry is the model TF and YFP is the protein product of the target gene. Decoys sites are used to control the network size by simulating the demand of other target genes in the SIM motif. (**F**) Representation of the tunable parameter space detailed in this study. We can systematically tune the TF unbinding rate, number of decoys and protein degradation rate in the experimental system and adjust these parameters accordingly in simulations.

related genes (*Alon, 2006*). There are mounting examples, from diverse topics that range from metabolism (*Figure 1B*, *Zaslaver et al., 2004*), stress response (*Figure 1C*, *Friedman et al., 2005*; *Ronen et al., 2002*), development (*Arnone, 2002*; *Gaudet and Mango, 2002*; *Kalir et al., 2001*), and cancer (*Lorenzin et al., 2016*), where temporal ordering of gene expression in the motif naturally follows the functional order of the genes in the physiological pathway. Mechanistically, it is thought that this ordering is set through differential affinity for the TF amongst the various target genes in the motif (*Alon, 2006*), although in some experiments temporal ordering was not observed implying a dependence on physiology or another experimental detail that is yet unrecognized (*Gerosa et al., 2013*; *Schmidt et al., 2016*). Due to the broad importance of these motifs, a quantitative understanding of how SIM motifs can be encoded, designed and optimized, will be instrumental in gaining a deep and fundamental understanding of the spatial and temporal features of a diverse set of cellular phenomena.

To quantitatively explore the input-output relationship of the SIM motif, we use a synthetic biology approach that boils the motif down to its most basic components: an autoregulated TF gene, a sample target gene, and competing binding sites. Using *E. coli* as a model organism, we build this

motif in vivo. We use non-functional 'decoy' binding sites to exert competition for the TF and mimic the demand of the other genes in the motif (which will depend on the size of the network, *Figure 1D*; *Gillespie, 1977*; *Shen-Orr et al., 2002*). However, the demand for the TF could also stem from a litany of sources such as random non-functional sites in the genome (*Bakk and Metzler, 2004*; *Kemme et al., 2016*; *Lee and Maheshri, 2012*; *Mirny, 2010*) or non-DNA-based obstruction or localization effects that transiently interfere with a TFs ability to bind DNA. Because of the design, our results do not depend on the nature of the TF competition. SIM TFs typically exert the same regulatory role on all targets of the motif (*Shen-Orr et al., 2002*). As such, in this work, we will focus on a TF that is a negative regulator of its target genes and itself; this is the most common regulation strategy in *Escherichia coli* where roughly 60% of TF genes are autoregulated and almost 70% of those TFs negatively regulate their own expression (inset *Figure 1D*, *Shen-Orr et al., 2002*).

We use stochastic simulations of kinetic models (*Gillespie, 1977*; *Gillespie, 2007*; *Kaern et al., 2005*; *Shahrezaei and Swain, 2008*), to predict how the overall level of gene expression depends on parameters characterizing cellular environment such as TF-binding affinities and the number of competing binding sites. To test these predictions in vivo, we built a synthetic system with LacI as a model TF, and individually tune each of these parameters. Past work with LacI has demonstrated the ability to control with precision the regulatory function, binding affinity and TF copy number through basic sequence level manipulations (*Brewster et al., 2014*; *Choi et al., 2008*; *Garcia and Phillips, 2011*; *Jones et al., 2014*; *Kuhlman et al., 2007*; *Oehler et al., 1990*; *Razo-Mejia et al., 2018*); Here, we use that detailed knowledge to inform our simulations which then guide our experiments (and vice versa).

Our approach reveals that the presence of competing TF-binding sites can have counterintuitive effects on the mean expression levels of the TF and its target genes due to the opposing relationship between free TFs and total TFs (total TF is the sum of free TF and TF bound to promoters and decoy binding sites). Furthermore, we find that the TF and target gene experience quantitatively different levels of regulation in the same cell, and with the same regulatory sequence. We show that this regulatory asymmetry is sensitive to features such as the degradation rate, TF-binding affinity and the number of competing binding sites for the TF. The stochastic simulation makes accurate predictions of the asymmetry and its dependence on the parameters of the model that we confirm through in vivo measurements. Interestingly, regulatory asymmetry is not captured by a simple deterministic model which is based on translating the stochastic reactions to kinetic rates through mass action equilibrium kinetics (which have been shown to accurately predict target gene expression in other studies [*Brewster et al., 2014*; *Garcia and Phillips, 2011*; *Garcia et al., 2012*; *Jones et al., 2014*; *Razo-Mejia et al., 2018*]). In fact, this deterministic model fails to accurately predict expression of either gene. A revised deterministic model, which explicitly allows for different microenvironments in each 'regulatory state', predicts asymmetry, although it still does not recover quantitative agreement with stochastic simulations.

## Results

### Matching molecular biology with simulation methodology

We use a combination of theory and experimental in vivo measurements on engineered *E. coli* strains to study the interplay between TF gene, target gene, and additional binding sites of a negative autoregulatory SIM network motif. The basic regulatory system is outlined in *Figure 1E*. We use a stochastic model of the SIM motif to explore how the expression of the TF gene and one target gene depends on parameters such as TF-binding affinity and number of other binding sites in the network (here modeled and controlled through competing, non-regulatory decoy sites [*Burger et al., 2010*]). In this model, the TF gene and target gene can be independently bound by a free TF to shut off gene expression until the TF unbinds. The two genes (TF-encoding and target) compete with decoy binding sites which can also bind free TFs. Each free TF can bind any open operator site with equal probability (set by the binding rate). The unbinding rate can be set individually for the TF gene, target gene and decoy sites and is related to the specific base pair identity of the bound operator site (*Kinney et al., 2010*; *Maerkl and Quake, 2007*; *Stormo, 2000*; *Weirauch et al., 2013*). We employ stochastic simulations to make specific predictions for how the expression level of the TF and target genes depend on the various parameters of the model.

Furthermore, we translate these stochastic processes into a deterministic ODE model using equilibrium mass action kinetics (see Appendix 6: Deterministic solution). A thorough discussion on how we chose the kinetic parameters of our model is presented in the Materials and methods section.

In experiments, the corresponding system is constructed with an integrated copy of both the TF (LacI-mCherry) and target gene (YFP) with expression of both genes controlled by identical promoters with a single LacI-binding site centered at +11 relative to their transcription start sites (*Brewster et al., 2014*; *Garcia and Phillips, 2011*). As demonstrated in *Figure 1F*, decoy binding sites are added by introducing a plasmid with an array of TF-binding sites (between 0 to 5 sites per plasmid) enabling control of up to roughly 300 binding sites per cell (for average plasmid copy number measured by qPCR, see Materials and methods and *Appendix 3—figure 1*). TF unbinding rate is controlled by changing the sequence identity of the operator sites; the binding sequence assessed in this study include (in order of increasing affinity) O2, O1 and Oid. The decoy binding site arrays are constructed using the Oid operator site. We quantify regulation through measurements of fold-change (FC) in expression which is defined as the expression level of a gene in a given condition (typically a specific number of decoy binding sites) divided by the expression of that gene when it is unregulated. For the target gene, we can always measure unregulated expression simply by measuring expression in a LacI knockout strain. However, it is challenging to measure unregulated expression for the autoregulated gene. For autoregulation, this unregulated expression can be measured by exchanging the TF-binding site with a mutated non-binding version of the site. For O1 there is a mutated sequence (NoO1v1, *Oehler et al., 1994*) that we have shown relieves repression of the target gene comparable to a strain expressing no TF (see *Appendix 4—figure 1A*), which allows us to calculate fold-change even for the autorepressed gene. Despite testing many different mutated sites and strategies, we could not find a corresponding sequence for O2 and Oid so we focus primarily on studying a TF gene regulated by O1 (see Appendix 4: Constitutive values for autoregulatory gene, for more discussion).

## Decoy sites increase expression of the auto-repressed gene and its targets

We first investigate the negatively regulated SIM motif where the TF and target gene have identical promoters and TF binding sites (O1) and the number of (identical) competing binding sites are varied systematically (schematically shown in *Figure 1E,F*). Simulation and experimental data for Fold-change of the TF gene as a function of number of decoys is shown in *Figure 2A* as red lines (simulation) and red points (experiments). We find that increasing the number of decoy sites increases the expression of the auto-repressed TF gene monotonically. To interpret why the TF level increases, in *Figure 2B* we plot the number of 'free' TFs in our simulation (defined as TFs not bound to an operator site) as a function of decoy site number. The solid line demonstrates that on average, despite the increased average number of TFs in the cell, the number of unbound TFs decreases as the number of competing binding sites increases (*Nevozhay et al., 2009*). Therefore, because the number of available repressors decreases, the overall level of repression also decreases and thus the mean expression of the TF gene rises.

Now we consider the effect of competition on the expression of SIM target genes. We measure our system with O1 as the regulatory binding site for both TF and target genes. In *Figure 2A*, the expression of the target gene is shown as blue points (experiments) and blue lines (simulation) for the SIM motif with different numbers of decoy TF-binding sites (from 0 sites up to five per plasmid). Just as in the case of the TF gene, we once again see that the expression of the target gene increases as more decoy binding sites are added even though the total number of TFs is also increasing (red points and line). Qualitatively, we expected this result since the free TF number is expected to decrease (*Figure 2B*) and, in turn, the expression of any gene targeted by the autoregulated repressing TF will increase. While the mechanism is more obvious in this controlled system, it is important to note that this is a case where more repressors correlate with more expression of the repressed gene. It is easy to see how this relationship could be misinterpreted as activation in more complex in vivo system if the competition level of the TF is (advertently or otherwise) altered in experiments.

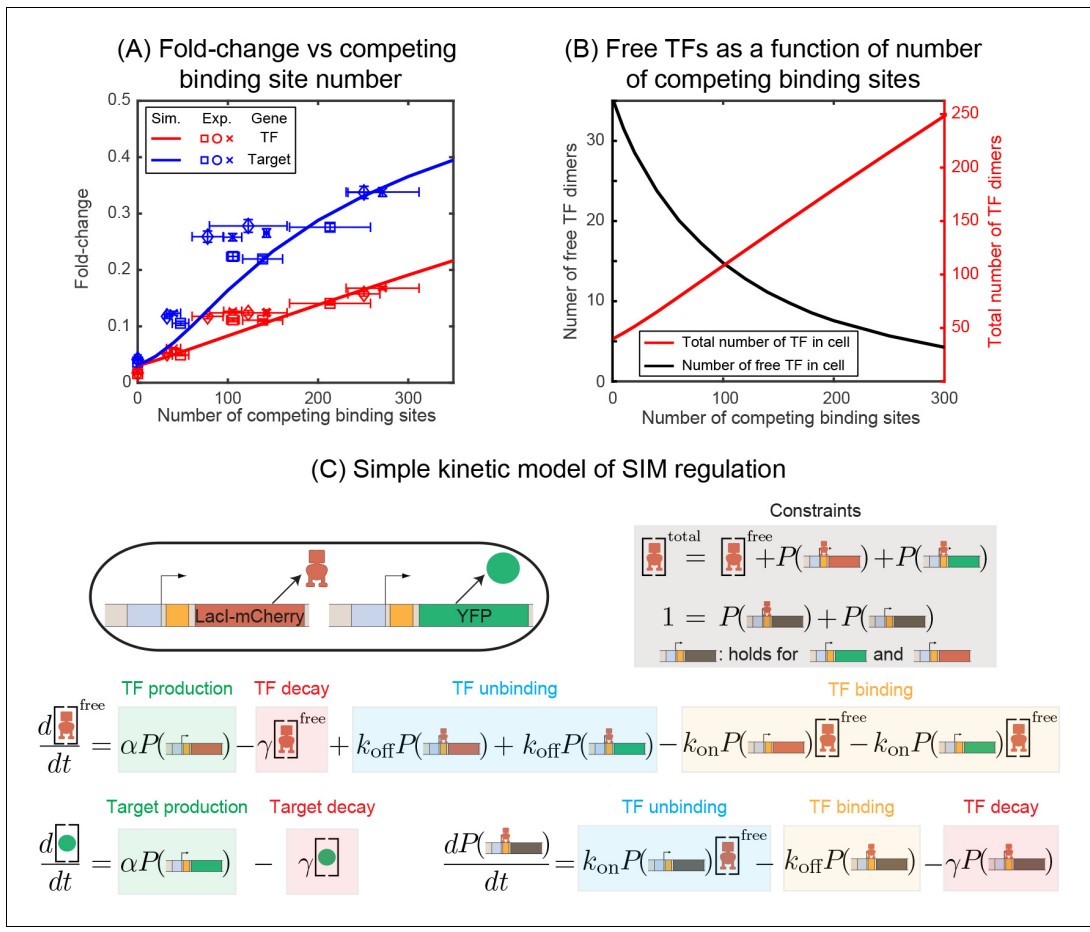

**Figure 2.** Fold-change in target and TF genes with network size. (**A**) Fold-change in the expression level of both the autoregulated gene (red) and the TF's target gene (blue) as a function of the number of competing binding sites present. Simulation data is shown as solid curves. Different symbols represent independent biological replicates. Each data point in y-axis is the bootstrapped mean of individual decoy strains and the error bars represent the standard deviation of bootstrapped mean. Each data point in x-axis is the mean of three technical replicates and the error bar is the corresponding standard deviation. (**B**) Increasing the number of competing binding sites increases the expression of both the TF (red line) and target genes by lowering the overall number of free TFs (black line). (**C**) Simple kinetic model describing the SIM motif using mass action equilibrium kinetics. For compactness of the figure, the reactions involving the decoy binding sites, dimerization/dedimerization of TF monomers, and transcription steps are not shown. Full reactions of the model are described in Appendix 6.

## Asymmetry in gene regulation between TF and target genes

Quantitative inspection of *Figure 2A* reveals an interesting detail: Even when the regulatory region of the auto-repressed gene and the target gene are identical, we find that the expression (fold-change or FC) is higher for the target gene, raising the question of how two genes with identical promoters and regulatory binding sites in the same cell can have different regulation levels. In this data, both the TF gene and target gene are regulated by a single repressor-binding site (O1) immediately downstream of the promoter. This regulatory scheme is often referred to as 'simple repression' (*Bintu et al., 2005*; *Garcia and Phillips, 2011*; *Phillips et al., 2013*). Drawing our intuition from a simple deterministic model of regulation based on translating the stochastic reactions to kinetic rate equations (*Figure 2C* and Appendix 6: Deterministic solution), we find that regardless of the network architecture (autoregulation, constitutive TF production, number of competing sites, *etc.*), the fold-change of any gene is expected to follow a simple scaling relation,

$$\text{Fold} - \text{change} = \frac{1}{1 + R^*},$$
$$R^* = R_{\text{free}} \frac{k_{\text{on}}}{k_{\text{off}} + \gamma}.$$

where, $R_{\text{free}}$ is the number of free (unbound) TFs and $k_{\text{on}}/(k_{\text{off}} + \gamma)$ represents the affinity of the specific TF binding site in the thermodynamic framework (*Rydenfelt et al., 2014*). This calculation is applicable for both the TF and the target gene and would predict a 'symmetric' response for identical regulatory regions. This model performs well for this same promoter in a related system where the TF is induced or constitutively expressed and predicts the fold-change for a wide range of perturbations such as promoter strength, TF-binding site, induction condition and TF competition levels

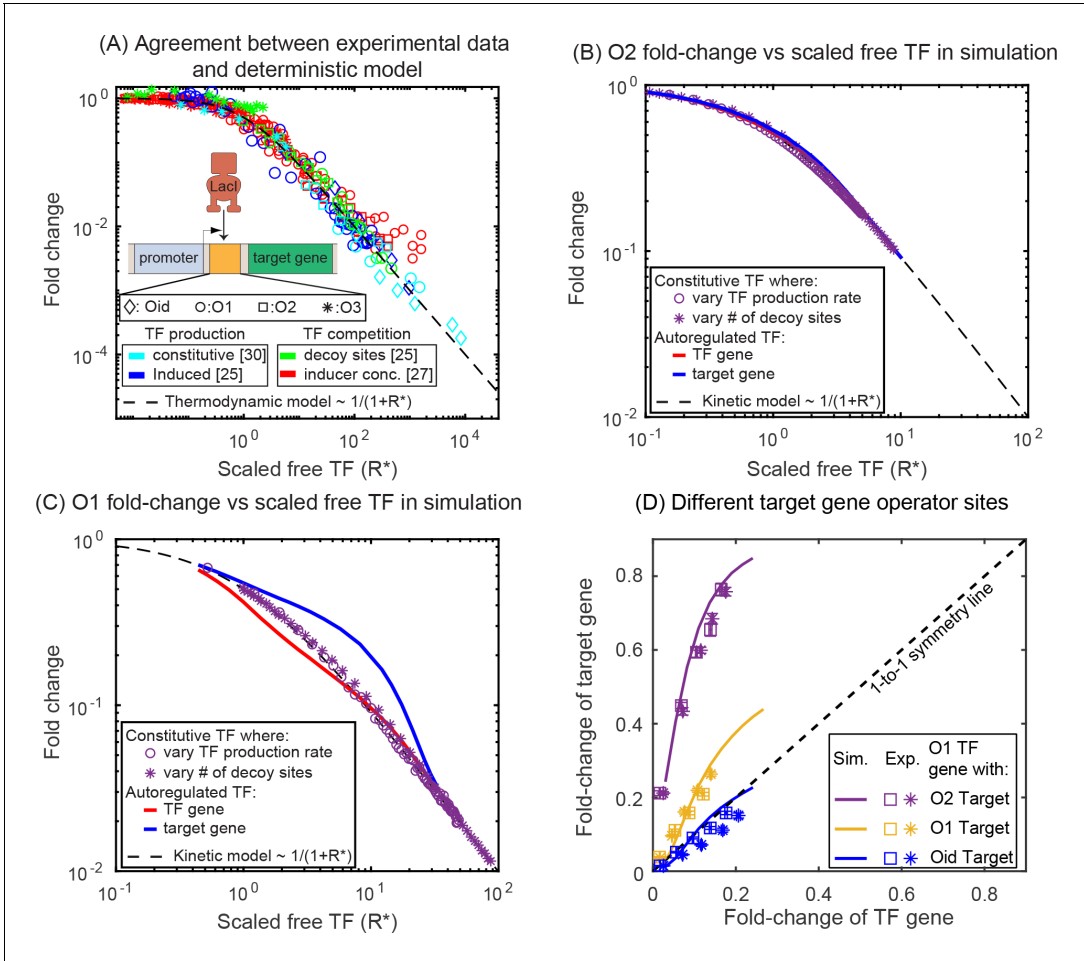

**Figure 3.** Comparison of SIM motif fold-change data to deterministic model predictions. (**A**) Fold-change vs scaled free TF in the thermodynamic model for a collection of simple repression data (open circles) where free TF is controlled through a diverse range of mechanisms. The data collapse to the deterministic model predictions (dashed curve). (**B–C**) Fold-change vs scaled free TF in simulations using the actual free TF obtained from simulation. The data for a constitutive expressed TF where free TF is varied by changing TF production rate (purple circles) or number of decoy sites (purple stars) collapses to the deterministic solution, however, the regulation of genes in the SIM motif (target: red line, TF gene: blue line) both diverge from the deterministic solution in opposing ways, giving rise not only to asymmetry but also a disagreement with deterministic modeling for both genes. (**D**) Fold-change in the target gene versus fold-change in the TF gene. Each data point is the bootstrapped mean of fold-change in TF and target expression across hundreds of cells with a given number of competing binding sites and error bars represent the standard deviation of the bootstrapped mean. Different symbols represent independent biological replicates. In all cases, the TF gene is regulated by an O1 binding sites, whereas the target is regulated by (in order of weakest binding to strongest binding): O2 (purple), O1 (yellow) or Oid (blue). Simulation data is shown as solid curves.

The online version of this article includes the following source data for figure 3:

**Source data 1.** Summary of results for fold-change in TF and target gene.

are tuned (data accumulated in *Figure 3A*, adapted from *Phillips et al., 2019*). However, it has been shown that the regulation of an autorepressed gene can diverge from this prediction (*Hahl and Kremling, 2016*; *Hornos et al., 2005*; *Milias-Argeitis et al., 2015*). In *Figure 3*, we show simulation data for the fold-change versus number of scaled-free TFs ($R^*$) for the autoregulatory gene (red line) and its target gene (blue line) with O1 (*Figure 3C*) or O2 (*Figure 3B*) binding sites, where we are changing the number of free TFs by tuning the number of competing-binding sites. In each plot, we also show simulations for the fold-change of a single target gene with a TF undergoing constitutive (constant in time) expression where the TF is controlled by either changing the expression level of the TF (purple stars) or adding competing-binding sites while maintaining a set constitutive expression level (purple circles). In both cases, where TFs are made constitutively, the simulation data agrees well with the deterministic model predictions. However, for the autoregulatory circuits, we find that for strong binding sites (O1) neither the target nor the TF gene follow the deterministic solution (black dashed line). In this case, the asymmetry occurs with the TF gene being more repressed and the target gene less repressed than expected.

Since 'free TF concentration' is not readily available in experiments, we demonstrate asymmetry in experimental results explicitly in , where we plot the fold-change of the target gene against fold-change of the TF gene. In this figure, the data points are derived from measurements made in six different competition levels (from 0 to 5 decoy binding sites per plasmid). Each data point represents the average expression level of each gene for a given number of competing binding sites. The lines represent results from the stochastic simulations where we systematically vary competition levels by introducing decoy binding sites and the fold-change of both the TF and target gene are calculated. The simple deterministic model prediction that identical promoters (yellow data, *Figure 3D*) should experience identical levels of regulation (see *Appendix 6—figure 1C*, *Sanchez et al., 2011*) would cause the data to fall on the black dashed one-to-one line. However, for both simulations and experiments of this system the TF gene is clearly more strongly regulated than the target gene subject to identical regulatory sequences.

To examine the extent of asymmetry in this system, we adjust the target binding site to be of higher affinity (Oid, blue lines and data points in *Figure 3D*) or weaker (O2, purple lines and data points in *Figure 3D*). Clearly, this should change the symmetry of the regulation, after all the TF-binding sites on the promoters are now different and symmetry is no longer to be expected. The experiments and simulations once again agree well. However, when Oid regulates the target gene and O1 regulates the TF gene, the regulation is now roughly symmetric despite the target gene having a much stronger binding site; in this case, the size of the inherent regulatory asymmetry effect is on par with altering the binding site to a stronger operator resulting in symmetric overall regulation of the genes.

## Mechanism of asymmetric gene regulation

The difference in expression between the TF and its target can be understood by studying the TF-operator occupancy for each gene, drawn schematically in *Figure 4A*. This cartoon shows the four possible promoter occupancy states of the system: (1) both genes unbound by TF, (2) target gene bound by TF, TF gene unbound, (3) TF gene bound by TF, target gene unbound, and (4) both genes bound by TF. It should be clear that state 1 and state 4 cannot be the cause of asymmetry; both genes are either fully on (state 1) or fully off (state 4). As such, the asymmetry must originate from differences in states 2 and 3. In state 2, the TF gene is 'on' while the target gene is fully repressed and in state 3 the opposite is true. Since we know that the asymmetry appears as more regulation of the TF gene than the target gene, then it must be the case that the system spends less time in state 2 than in state 3. There are two paths to exit either of these states: unbinding of the TF from the bound operator or binding of the TF to the free operator. Since unbinding rate of a TF is identical for both promoters in our model, the asymmetry must originate from differences in binding of free TF in state 2 and in state 3; specifically state 2 must have an (on average) higher concentration of TF than state 3. This makes sense since the system is still making TF in state 2, while production of TF is shut off in state 3. *Figure 4B* validates this interpretation as we can see that state 2 has on average more free TFs than state 3, and as a result, the system spends less time in state 2 than in state 3 in our simulations. As such, the asymmetry comes from the fact that the two genes, despite being in the same cell and experiencing the same average intracellular TF concentrations, are exposed to systematically different concentrations of TF when the TF and target gene are in their respective 'active'

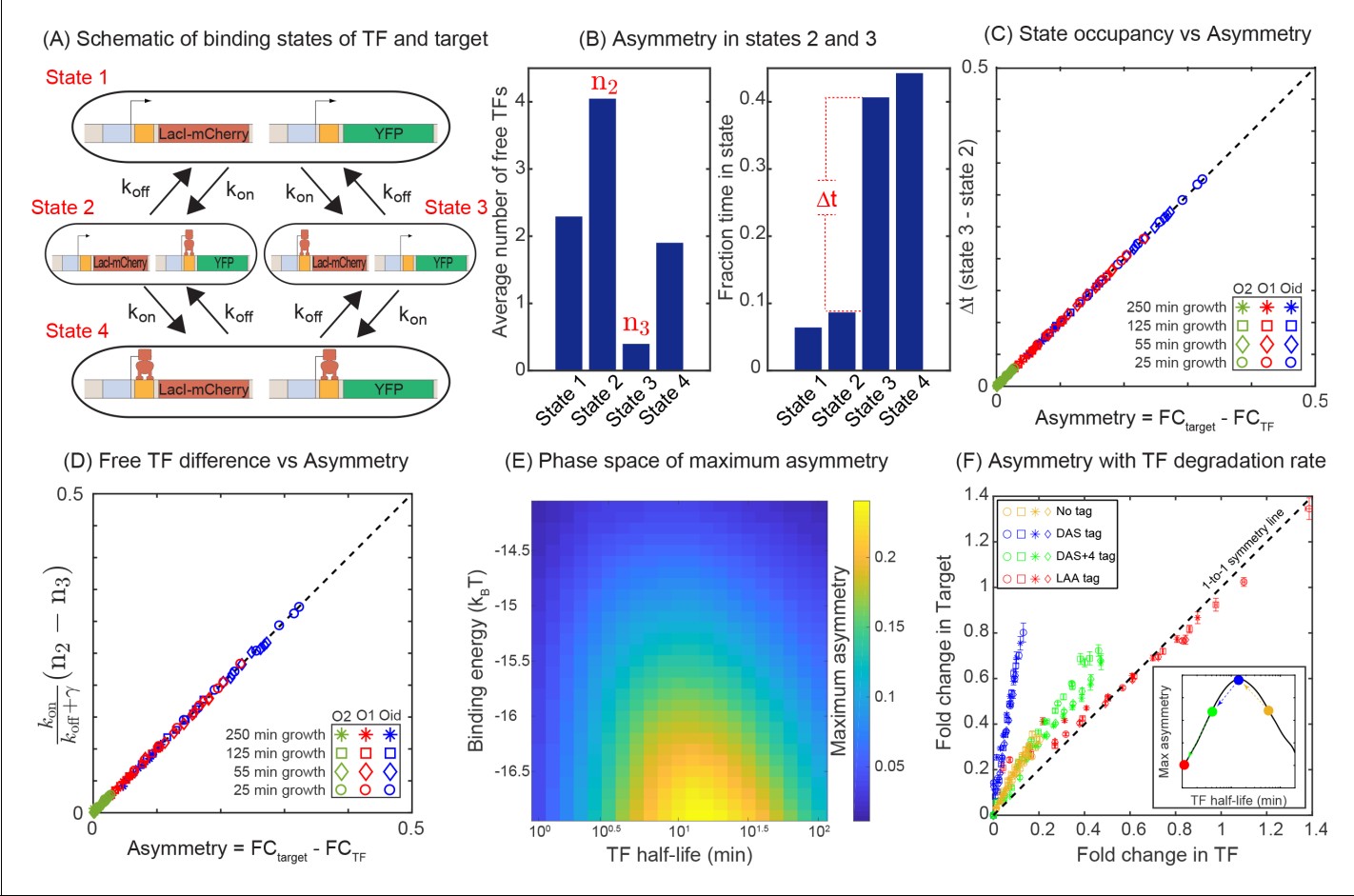

**Figure 4.** Mechanism of regulatory asymmetry. (**A**) Schematic of the TF-operator occupancy with their corresponding transition rates. The $k_{on}$ for transition from state 1 to state 2 or state 3 will be identical and hence cannot account for the asymmetry. State 2 and state 3 on the other hand, will encounter a difference in the free TF concentration and hence the $k_{on}$ for transition from one of these states to state 4 will be different; thus, accounting for the asymmetry in expression between the TF and the target. (**B**) Plot showing the average number of free TFs in different states and fraction of time cells spends in each of the given state in the simulation. (**C**) Plot showing asymmetry as a function of fractional time difference between state 2 and state 3. (**D**) Plot showing asymmetry as a function of difference in free TF concentration between state 3 and state 2. (**E**) Heat map showing the phase space of maximum asymmetry as a function of binding affinity for the TF and its half-life. (**F**) Tuning the TF degradation rate influences the extent of asymmetry observed in the SIM module. Yellow points corresponds to the system with no degradation tags; Blue points corresponds to degradation by a 'weak' or 'slow' tag (DAS tag with a rate of 0.00063 per min per enzyme); Green points corresponds to a slightly faster tag (DAS + 4 with a rate of 0.0011 per min per enzyme ); Red points corresponds to a very fast tag (LAA tag with a rate of 0.21 per min per enzyme ). Different symbols represent independent biological replicates.

The online version of this article includes the following source data for figure 4:

**Source data 1.** Summary of results for strains expressing TFs with different degradation rates.

states. To quantify regulatory asymmetry, we define asymmetry as the difference in fold-change of the target and the fold-change of the TF gene (asymmetry $= \mathrm{FC}_{target} - \mathrm{FC}_{TF}$). Using the chemical master equation (CME) approach, we find that the asymmetry is exactly equal to the difference in time spent in state 3 and state 2, for any condition or parameter choice (*Figure 4C* and Appendix 9 *Equation A9-9*: CME for minimal model). Furthermore, the asymmetry can be written as the difference of TF concentration in state 2 and state 3 and is given by

$$\mathrm{Asymmetry} = \frac{k_{on}}{k_{off} + \gamma}(n_2 - n_3),$$

where, $n_2$ and $n_3$ are the TF concentrations in state 2 and state 3, respectively. In *Figure 4D*, we show that the asymmetry obtained using the difference in TF concentration precisely match with the

asymmetry calculated from the fold-change expression. However, it is important to note that this is not a complete analytic solution for asymmetry because $n_2$ and $n_3$ are unsolved functions of the model parameters.

The asymmetry in the expression of TF and target genes stems from systematically differential TF concentration in the states when the TF gene is occupied (and target gene is expressing) and when the target gene is occupied (and the TF gene is expressing). The general approach of ODEs outlined above (*Figure 2C*) does not account for this differential TF concentration and hence shows no asymmetry. Armed with the knowledge that individual states have this systematic TF difference, we can rewrite the basic deterministic model where we instead keep track individually of each state and the specific TF concentration of that state using the same equilibrium mass action kinetic approach (details in Appendix 10: Modified ODEs for the minimal model). Like the stochastic CMEs, the modified ODEs predict that the asymmetry arises from the difference in the TF concentrations in different states and solely depends on the difference in time spent in state 3 (only target gene occupied) and state 2 (only TF gene occupied). Although we find the modified deterministic model can predict asymmetry, it still does not quantitatively agree with the results of stochastic modeling due to the deterministic model not accounting for variability in TF number in each state (see Appendix 10: Modified ODEs for the minimal model). As a result, in the following sections, we will compare our experiments to stochastic simulations based on the full CME formalism.

## Dependence of regulatory asymmetry on TF degradation and binding affinity

According to the above-proposed mechanism, the regulatory asymmetry stems from differences in the cellular TF concentration when the TF is bound to the target versus when it is bound to the autoregulatory gene, as such we expect that binding affinity will play a central role in setting asymmetry levels. This is also evident from *Figure 3B,C* where we find that the deviation of the expression of both TF and target gene is more prominent for a strong binding site (Oid or O1) compared to a weaker binding site (O2). Furthermore, there are many parameters associated with the production and decay of TF and target mRNA and protein which could also influence the asymmetry. To reveal which (if any) of these parameters is important to asymmetry, we calculate the maximum asymmetry (the maximum value of asymmetry found as competing site number is controlled, *Appendix 7—figure 1A*) using simulation as these production and degradation parameters are tuned. First, we find that tuning the rates of target gene production and decay has no effect on asymmetry (*Appendix 7—figure 1B* and *Appendix 11—figure 1B*). On the other hand, for TF production and decay each parameter has some effect on asymmetry. However, we find that the biggest driver of asymmetry in this set of parameters is the protein degradation rate (*Appendix 7—figure 1B*). As such, we focus on two crucial parameters that control the asymmetry: TF-binding affinity and TF degradation rate. In *Figure 4E*, we show a heat map of the maximum asymmetry as a function of the rate of protein degradation and binding affinity of the TF. We see from this figure that strong binding produces enhanced asymmetry, but the degradation rate displays an interesting intermediate maximum in asymmetry — degradation that is too fast, or too slow will not show asymmetry, but a maximum asymmetry is expected for TF lifetimes between 10 and 100 min. Crucially, this maximum coincides with typical doubling time of *E. coli* (which sets the TF half-life [*Marr, 1991*; *Neidhardt and Curtiss, 1996*]) and thus regulatory asymmetry in this motif is most relevant in common physiological conditions.

The non-monotonic behavior of asymmetry with degradation rate of TF can be explained by the TF-promoter occupancy (alternatively, residence time) of the TF and the target gene. Analytically, the asymmetry is given by the difference of occupancy of state 2 and state 3 (Appendix 9 *Equation A9-7*: CME for minimal model). For slow degradation, the number of TFs in a cell is high, favoring the transition to state 4 very quickly, thereby reducing the residence times of both state 2 and 3. On the other extreme, when degradation is fast, the TF number is too low for the cell to be in the state 2 or 3; the cell spends most of the time in state 1. In both the cases, the difference of residence times between state 2 and state 3 is low and hence the asymmetry is small. In the intermediate regime of degradation, the number of TFs is optimum to maximize the difference between residence times in state 2 and 3, which leads to maximum asymmetry.

To experimentally test the theory predictions for the role of TF degradation in setting regulatory asymmetry, we introduced several ssrA degradation tags to the LacI in our experiments

(*McGinness et al., 2006*). The data, shown in *Figure 4F* includes degradation by a 'weak' or 'slow' tag (DAS with a rate of 0.00063 per minute per enzyme [*McGinness et al., 2007*], blue points), a slightly faster tag (DAS + 4 with a rate of 0.0011 per minute per enzyme [*McGinness et al., 2007*], green points) and a very fast tag (LAA tag with a rate of 0.21 per minute per enzyme [*McGinness et al., 2007*], red points) . In addition, the data without a tag is shown as yellow points. Here, we see that the slowest tag (blue points) introduces strong asymmetry. However, for the next fastest tag (green points) we see a significant decrease in asymmetry and the level of regulatory asymmetry is similar to what is seen in the absence of tags (yellow points). Finally, the fastest tag (red points) shows no asymmetry at all. It is worth pointing out that the qualitative order of degradation rates in these experiments can be inferred from how far the data 'reaches', faster degradation will lead to higher overall fold-changes for a given competition level. Importantly, controlling the protein degradation rate through this synthetic tool agrees with our model predictions, although the actual in vivo protein degradation rates are difficult to estimate from tag sequence alone, the asymmetry follows the expected trends based on the known (and observed) effectiveness of each tag (see schematic inset *Figure 4F*).

In the absence of targeted degradation, the degradation rate of most protein in *E. coli*, is naturally set by the growth rate. According to the model predictions in *Figure 4E*, the asymmetry should be highest for fast growing cells (roughly 20 min division rate for our growth conditions which is well below the degradation rate for peak asymmetry ~10 min, *Figure 4F*) and decrease (or vanish) for very slow growing cells. To test this, we take the system with O1 regulatory binding sites on both the target and the TF promoter (yellow data in *Figure 3D* grown in M9 + glucose, 55 min doubling time) and grow in a range of doubling times between 22 min (rich defined media) up to 215 min (M9 + acetate) (see *Appendix 2—figure 1A*). Importantly, when we change the growth rate, other rates such as the transcription and translation rates will also be impacted (*Bremer and Dennis, 2008*; *Klumpp et al., 2009*), while these parameters will change the quantitative values of the asymmetry curve, the qualitative ordering and features of the asymmetry are not expected to be impacted (see *Appendix 11—figure 1C*). The data for these growth conditions is shown in *Figure 5A*. As predicted, faster growing cells show more regulatory asymmetry and slower growing cells show little-to-no regulatory asymmetry. We also test the role of growth rate in asymmetric regulation when O2 (a lower affinity site) and Oid (a higher affinity site) are used as the regulatory-binding sites instead of O1. This data is shown in *Figure 5B* (O2) and 5C (Oid). As discussed above, we could not find a suitable mutant for O2 and Oid that both relieved regulation from LacI and completely restored the expression of target gene (see Appendix 4: Constitutive values for autoregulatory gene.). This means we cannot explicitly measure the 1–1 correlation between the two axes in our data when using O2 or Oid for the TF gene. To this end, we find this correspondence by fitting the glucose data to our simulation of the same system and use that value to normalize all other growth rates for that operator. Despite this complication, it is clear that O2 regulation is symmetric at all studied growth rates while Oid regulation is asymmetric for all growth rates with faster growth rates appearing more asymmetric.

Importantly, the regulatory asymmetry is not due to a small population of outliers, bimodality or any other 'rare' phenotype. In *Figure 5D*, we show a histogram of single cell asymmetry values (defined as asymmetry = $\mathrm{FC_{Target}} - \mathrm{FC_{TF}}$) for each condition. As can be seen, expression in each media condition are roughly symmetric for most cells at the lowest competition levels (top panel). However, as competition levels are increased, the fast-growing conditions shift to higher asymmetry levels; strikingly at the highest growth rate almost every single cell is expressing target at a higher level than TF (bottom panel).

## Discussion

The single-input module (SIM) is a prevalent regulation strategy in both bacteria (*Ma et al., 2004*; *Shen-Orr et al., 2002*) and higher organisms (*Lee et al., 2002*; *Segal et al., 2003*; *Yu et al., 2003*). While the role of TF autoregulation (positive and negative) has been extensively studied (*Acar et al., 2008*; *Assaf et al., 2011*; *Becskei and Serrano, 2000*; *Ochab-Marcinek et al., 2017*; *Rodrigo et al., 2016*; *Rosenfeld et al., 2002*; *Savageau, 1975*; *Semsey et al., 2009*), the focus here is on the combined influence of an autoregulated TF and its target genes and how the shared need for that TF influences the quantitative features of its regulatory behaviors. We find that there is

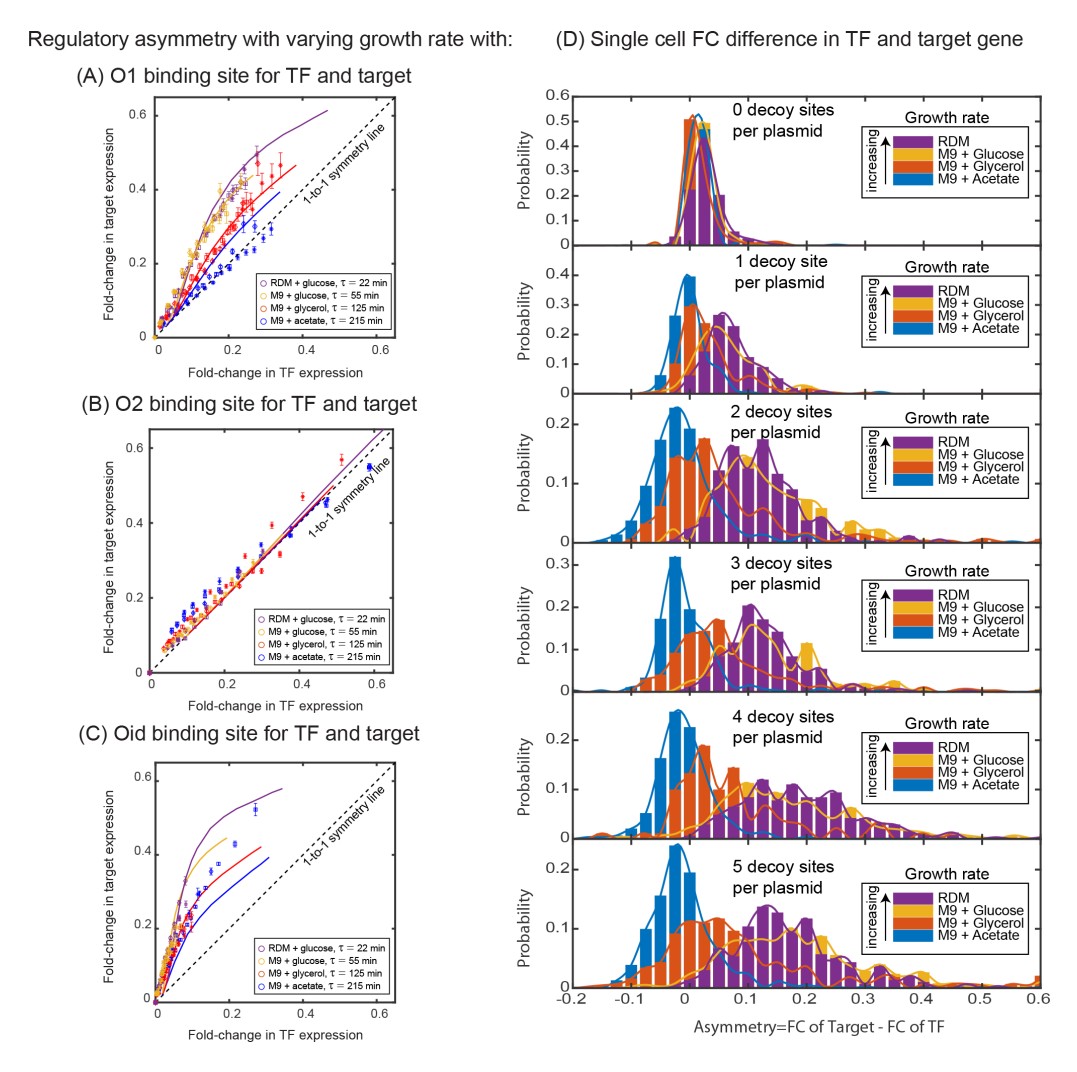

**Figure 5.** Dependence of regulatory asymmetry on growth rate. Measurement of asymmetry in different media as a function of TF binding energy: O1 (**A**), O2 (**B**), Oid (**C**). The division time (τ) is varied between 22 min up to 215 min. (**A**) For O1, the asymmetry decreases with slower division rates and agrees well with the simulation predictions. (**B**) For the weak O2 site, no asymmetry is seen at any growth rate. (**C**) For the strongest site, Oid asymmetry is present at every growth rate although the magnitude of asymmetry still orders roughly by growth rate. Different symbols represent independent biological replicates and simulation data are shown as solid curves. (**D**) Histograms of single-cell asymmetry in expression of the TF and target gene regulated by O1 binding site in these four growth rates. Solid lines represent the interpolated distributions for better visualization of the histograms. Panels from top to bottom represent increasing the level of competition for the TF.

The online version of this article includes the following source data for figure 5:

**Source data 1.** Summary of results for strains grown under different physiological conditions.

a fundamental asymmetry in gene regulation that can occur in the SIM regulatory motif. This asymmetry is not related to distinctions in the biological processes or an unexpected difference in our in vivo experiment, but rather an inherent asymmetry originating from the way the motif itself is wired. Although two identical promoters are in the same cell with the same average protein concentrations, they experience distinct regulatory environments. This is particularly relevant for the SIM motif because the primary function of the motif, organizing and coordinating gene expression patterns, operates on the premise of differential affinities amongst target genes; here we have shown that the TF gene has an inherent 'affinity advantage' due to being exposed to systematically higher TF concentrations than its target genes. This implies that the TF gene will respond 'earlier' than expected based on the raw affinity of its binding site and may necessitate weaker sites on autoregulating TF genes in order to achieve similar timing in expression compared to its targets. This may also shed

light on the discrepancies in Arg pathway timing between different experiments which have used plasmid reporters (essentially changing network size) or different physiological growth conditions; the asymmetry is critically sensitive to both of these features. Although, here we are using *E. coli* as a model organism where it is easy to build and manipulate these regulatory motifs, we expect this phenomenon to apply broadly to other regulatory systems.

Regulatory asymmetry is intrinsic to the negative SIM motif even in the absence of decoys, but it can be greatly exacerbated by competing TF-binding sites. Due to the promiscuous nature of TF binding, this highlights the importance of considering not just the 'closed' system of a TF and a given target but also the impact of other binding sites (or inactivating interactions) for the TF in predicting regulation as well as the regulatory motif at play in the system. In our system, the magnitude of the asymmetry is enough to compensate for swapping the wild-type proximal O1 LacI binding site on the target gene with the 'ideal' operator Oid.

The cause of this asymmetry is a systematic difference in the TF concentration when the TF gene is active compared to when the target gene is active. As such, asymmetry is magnified by anything that enhances this concentration difference. Here, we have identified TF-binding affinity and TF degradation rate (controlled both directly and through modulating growth rate) as primary drivers of asymmetry in this motif. Although the relationship between growth rate and expression levels is well established (*Bremer and Dennis, 2008*; *Klumpp and Hwa, 2008*; *Klumpp et al., 2009*; *Scott et al., 2010*; *Volkmer and Heinemann, 2011*), effects such as this add a layer of complexity to this relationship.

In studies of quantitative gene regulation, the typical goal is to predict the output of a gene based on the regulatory composition of that gene's promoter and the number and identity of regulatory proteins. This work clearly presents a challenge for the drive to 'read' and predict regulation levels from the promoter DNA alone, in this case the regulatory motif is responsible for altering the observed regulation and must be considered as well. It has previously been demonstrated that features of a transcript can impact its regulation by effects such as targeted degradation, stabilization or posttranslational modification and regulation (*Schikora-Tamarit et al., 2018*), it is important to point out that regulatory asymmetry in this motif is a distinct phenomenon that does not operate through an enzymatic processes but rather is a fundamental feature of the network.

Finally, here we demonstrate regulatory asymmetry using a specific (but common) regulatory motif. The more general problem of quantifying the role of asymmetry in other network motifs may be an important step in expanding the predictive power of models based on single genes. The broader point that specific genes can be exposed to systematically different levels of regulatory TFs even in the absence of specific cellular mechanisms such as cytoplasmic compartmentalization, protein localization or DNA accessibility is likely more generally relevant. Understanding and quantifying these mechanisms can be an important piece towards improving our ability to predict and design gene regulatory circuits.

## Materials and methods

**Key resources table**

| Reagent type (species) or resource | Designation | Source or reference | Identifiers | Additional information |
|---|---|---|---|---|
| Gene (*E. coli*) | *ybcN<>25XX+11-lacI-mcherry* | GeneBank | MT726947 | TF gene; XX can be O1, O2 or Oid operator |
| Gene(*E. coli*) | *galK<>3*5XX+11-yfp* | GeneBank | MT726948 | Target gene; XX can be O1,O2 or Oid operator |
| Strain, strain background (*E. coli*) | *E. coli* MG1655 | Lab stock | CGSC#6300 | Wild type |
| Strain, strain background (*E. coli*) | HG105 | *Garcia and Phillips, 2011* | | *E. coli* MG1655 with *lac* operon deleted |

*Continued on next page*

*Continued*

| Reagent type (species) or resource | Designation | Source or reference | Identifiers | Additional information |
|---|---|---|---|---|
| Strain, strain background (*E. coli*) | HG105 Δ*sspB* | This study | | *E. coli* HG105 with *sspB* gene deleted |
| Other | M9 minimal media | BDDiagnostics | DF0485-17 | Commercial media |
| Other | Rich defined media | Teknova | #M2105 | Commercial media |
| Software, algorithm | Matlab code | Schnitzcells **Rosenfeld et al., 2005** | | |
| Other | C code for simulations | GitHub link This study | | |

## Bacterial strains

All strains used in this study are constructed from the parent strain *E. coli* HG105 which is MG1655 with the *lac* operon deleted (MG1655 Δ*lacIZYA*). Auto-regulated TF (*lacI-mCherry*) is expressed from the *ybcN* locus and the TF-repressed target (*yfp*) is expressed from the *galK* locus with identical promoter sequence for both the TF and the target. Decoys are introduced on the pZE plasmid. In order to tune the degradation rate of the TF, three different ssrA tags were added to the C-terminus of the LacI-mCherry fusion protein. The tags used in this study are wild-type LAA tag (AANDENYA-LAA), DAS tag (AANDENYADAS) and DAS + 4 tag (AANDENYSENYADAS) (*McGinness et al., 2006*). For protein degradation tag experiments with LacI-mCherry fusion protein, HG105 with ΔsspB knockout is used as a parent strain to substantially moderate the protein degradation rate. It is also noteworthy that deletion of *sspB* gene did not affect the growth rate in any of the strains tested. Primers used in this study are listed in *Table 1*.

**Table 1.** Primers used in this study are listed below.
Primers for the chromosomal integration of TF and the target are the same as described in *Brewster et al., 2014*. Primers to mutate the binding sites from O1 to Oid, O2 or NoO1V1 is listed below with the binding sites in blue. Primers to introduce the degradation tags to LacI mCherry fusion protein is listed below with tag sequence in red.

**Mutagenesis primer**

| | |
|---|---|
| Oid_mutagenesis_FP | CCGGCTCGTATAATGTGTGGAATTGTGAGCGCTCACAATTGAATTCATTAAAGAG |
| Oid_mutagenesis_RP | CTCTTTAATGAATTCAATTGTGAGCGCTCACAATTCCACACATTATACGAGCCGG |
| O2_mutagenesis_FP | GTGAGCGAGTAACAACCGAATTCATTAAAGAGGAGAAAGGTAC |
| O2_mutagenesis_RP | TTGTTACTCGCTCACATTTCCACACATTATACGAGCC |
| NoO1V1_mutagenesis_FP | GATTGTTAGCGGAGAAGAATTGAATTCATTAAAGAGGAGAAAGGTACC |
| NoO1V1_mutagenesis_RP | AATTCTTCTCCGCTAACAATCCCACACATTATACGAGCCGGAAG |
| Primers to introduce tags | |
| ssrA_WT_FP | GCAGCAAACGACGAAAACTACGCTTTAGCAGCTTAAGCTTAATTAGCTGAGTCTAGAGGC |
| ssrA_WT_RP | AGCTGCTAAAGCGTAGTTTTCGTCGTTTGCTGCTTTGTACAGCTCATCCATGC |
| DAS_FP | CAGCAAACGACGAAAACTACGCTGATGCATCTTAAGCTTAATTAGCTGAGTCTAGAGGC |
| DAS_RP | AGATGCATCAGCGTAGTTTTCGTCGTTTGCTGCTTTGTACAGCTCATCCATGCAGCTCATCCATGC |
| DASplus4_FP | GCAGCAAACGACGAAAACTACTCTGAAAATTATGCTGATGCATCTTAAGCTTAATTAGCTGAGTCTAGAGGC |
| DASplus4_RP | AGATGCATCAGCATAATTTTCAGAGTAGTTTTCGTCGTTTGCTGCTTTGTACAGCTCATCCATGC |
| qPCR primers | |
| qPCR_FP | GCATTTATCAGGGTTATTGTCTCAT |
| qPCR_RP | GGGAAATGTGCGCGGAAC |

## Microscopy

Bacterial cultures are grown overnight in 1 mL of LB in a 37°C incubator shaking at 250 rpm. Unless otherwise stated cultures grown overnight are diluted $2.5 \times 10^3$ fold to an initial OD of 0.002 into 1 mL of fresh M9 minimal media supplemented with 0.5% of one of the three different carbon sources (Glucose, Glycerol or Acetate) or in Rich Defined Media (RDM, Teknova #M2105), allowed to grow at 37°C until they reach an OD600 of 0.2 to 0.4 (0.1 for acetate) and harvested for microscopy. Cells are diluted 1:3 in 1X PBS (in order to obtain isolated cells in microscope images) and 1L is spotted on a 2% low melting agarose pad (Invitrogen #16520050) made with 1X PBS. Cells grown in RDM are cross-linked with paraformaldehyde before imaging to prevent shrinkage and osmotic shock to the cells. An automated fluorescent microscope (Nikon TI-E) with a heating chamber set at 37°C is used to record multiple fields per sample (between 8 and 12 unique fields of view) resulting in roughly 500 to 1000 individual cells per sample.

## qPCR measurements for average plasmid copy number

We performed qPCR measurements in order to quantify the average copy number of the pZE plasmid. Cells are grown as described for microscopic analysis and diluted 1:200 in Qiagen P1 lysis buffer and allowed to sit on ice. Meanwhile, cells are plated at 10–5 dilutions on fresh LB plates in order to determine the colony-forming units per mL (CFU/mL). 25 L of the lysate is diluted with 25 L of 1X PBS and allowed to sit for 5 min. The cells are then diluted 1:100 into 1X cut smart buffer from NEB. 20L of the mixture is incubated with 0.5 L of HindIII restriction enzyme for 30 min at 37°C followed by heat inactivation at 80°C for 20 min. The mixture is further diluted 1:10 and 4.2 L is used as a template in a 20 L qPCR reaction mixture. The pZE-1XOid plasmid is purified using the Qiagen Plasmid Medi Prep kit and quantified using the Qubit dsDNA assay kit. A standard curve is then prepared by diluting pZE-1XOid plasmid from $10^8$ copies down to 10 copies. The average copy number of the decoy plasmid per cell is computed by comparing the cT of the sample to the standard curve and dividing by the number of cells in the sample.

## Simulation methodology

To model the experiments and study the effect of decoy sites on the expression of a target gene regulated by a negatively autoregulated TF gene, we develop a simple model of the experimental system. In our model, the auto-regulatory gene produces a protein (*X*) which forms a TF dimer (*R*). We explicitly modeled TF as a dimer to incorporate the fact that LacI acts as a dimer in our experimental system (the LacI-mCherry construct lacks the tetramerization domain [*Kipper et al., 2018*]). Dimerization and de-dimerization steps occur at the rate $k_p$ and $k_m$, respectively. The TF binds to its own promoter ($P_{TF}$), to the promoter of the target gene ($P_{target}$), and to the decoy sites (*N*) with a constant rate $k_{on}$ per free TF per unit time. The off rate of the bound TF ($k_{off}$, the unbinding rate) depends on the sequence identity and can be different for different promoters. A bound TF unbinds from the promoters of the TF and target, and from the decoy sites at a rate $k_{off,TF}$, $k_{off,target}$, and $k_{off,decoy}$ per unit time, respectively. A TF-free promoter produces an mRNA at the rate β which is then translated into a protein at a rate α. The mRNA and the proteins are degraded at the rate $\gamma_m$ and γ, respectively. We assume that all proteins (free protein, TF bound to promoter and TF bound to decoy sites) degrades with the same rate. Typically, the proteins in *E. coli* are very stable with protein half-life greater than the cell cycle and the dominant contribution to degradation comes from the dilution due to cell division. The degradation rate is thus given by $\gamma = ln(2)/\tau_1 + ln(2)/\tau_2$, where $\tau_1$ and $\tau_2$ are protein half-life and cell division time, respectively. The set of reactions describing the model above are listed in *Appendix 6—figure 2A*.

We implement the simulations for stochastic reaction systems using Gillespie's algorithm (*Gillespie, 1977*) in C programming. Each simulation is run for sufficiently long time (~$10^6$ s) to reach a steady state. Typically, for the rates used in this paper the steady state is achieved in $10^5$ s or less (see *Appendix 6—figure 1* for a sample time trace). Data for steady state distributions (TF and target protein) are then recorded by sampling over time with a time interval ($T_S$) long enough for the slowest reaction to occur 20 times on average ($T_S$ = 20 over rate for slowest reaction). Mean protein numbers in steady state for fold-change are calculated using at least $10^5$ data points for each single run.

## Kinetic parameter estimation

To compare the results from experiments with our simulations, we are required to find values for the kinetic on and off rate of LacI for different operator sites (Oid, O1 and O2), the transcription and translation rates, mRNA degradation rate, and the growth rates in different media. We directly measure growth rate for different media in our experiment (see Appendix 2). The on and off rates are related to the binding energy ($\Delta\epsilon$) through,

$$\frac{k_{\mathrm{on}}}{k_{\mathrm{off}} + \gamma} = \frac{exp(-\Delta\epsilon)}{N_{\mathrm{ns}}}, \tag{1}$$

where $N_{\mathrm{ns}} \sim 5 \times 10^6$ bps is the number of non-specific binding sites in the genome (which we take as the total number of bases) (*Phillips et al., 2013*), $k_{\mathrm{on}}$ is the binding rate per free TF per unit time, $k_{\mathrm{off}}$ is the unbinding rate per unit time and $\gamma$ is the decay rate of the TF. Experimental measurements of $\Delta\epsilon$ have been reported in many repeated experiments (*Brewster et al., 2014*; *Garcia and Phillips, 2011*; *Razo-Mejia et al., 2018*) and thus we constrain our choice of $k_{\mathrm{on}}$ and $k_{\mathrm{off}}$ such that we obtain affinities consistent with these measurements. Taking one data set (O1 regulated TF and O1 regulated target grown in glucose), we use maximum likelihood analysis to obtain the rates by varying $k_{\mathrm{on}}$ in a range 0.0015–0.003 $s^{-1}$ (which sets the corresponding value of $k_{\mathrm{off}}$ to give $\Delta\epsilon_{\mathrm{O1}} = 15.3 k_{\mathrm{B}} T$) (*Elf et al., 2007*; *Bremer and Dennis, 2008*), $\gamma_{\mathrm{m}}^{-1}$ in a range of 30–90 s (*Yu et al., 2006*; *Bremer and Dennis, 2008*), $\beta$ in a range of 0.1–0.3 $s^{-1}$ (*Kennell and Riezman, 1977*), and choosing $\alpha$ (*Cai et al., 2006*) such that the constitutive number for the TF protein is in the range of 1000–2600; this parameter largely sets the 'range' of our fold-change vs fold-change curves and this range of $\alpha$ reproduces the experimental range we see in those curves for this data set.

We then use this same on rate to derive the relevant off rates for O2 and Oid using their binding energies ($\Delta\epsilon_{\mathrm{O2}} = 13.9 k_{\mathrm{B}} T$, $\Delta\epsilon_{\mathrm{Oid}} = 16.3 k_{\mathrm{B}} T$ ) and *Equation 1*. Interestingly, the binding affinity we measure for Oid is 0.7 $k_{\mathrm{B}} T$ weaker than has been previously reported but is consistent with measurements of Oid binding affinity in our lab. Using this method, we find the $k_{\mathrm{on}}$ to be 0.0015 per TF per second, which yields $k_{\mathrm{off}}$ to be, O1 = 0.0015 $s^{-1}$, O2 = 0.0167 $s^{-1}$ and Oid = 0.0004 $s^{-1}$, consistent with previous findings (*Elf et al., 2007*; *Hammar et al., 2014*; *Jones et al., 2014*; *Razo-Mejia et al., 2018*). All other rates are listed in *Table 2*. Importantly, this process is not meant to precisely determine the exact quantitative parameters of LacI binding, and it is not a formal fit, but rather an estimate that provides us with realistic prediction of regulation from our simulations using molecular parameters that are consistent with available direct kinetic measurements (*Chen et al., 2015*; *Elf et al., 2007*; *Sanchez et al., 2011*; *Yu et al., 2006*).

**Table 2.** Kinetic rates used in the simulations.

| Rates | Symbols | Value | Reference |
|---|---|---|---|
| Growth rate | $\ln 2/\gamma$ | 25 min (RDM) | Measured experimentally |
| | | 55 min (Glucose) | |
| | | 125 min (Glycerol) | |
| | | 225 min (Acetate) | |
| Binding of TF | $k_{\mathrm{on}}$ | 0.0015 TF$^{-1}$s$^{-1}$ | Obtained from fit |
| Unbinding of TF | $k_{\mathrm{off}}$ | 0.00042 s$^{-1}$ (Oid) | *Equation 1* |
| | | 0.00149 s$^{-1}$ (O1) | |
| | | 0.0167 s$^{-1}$ (O2) | |
| mRNA degradation | $\gamma_{\mathrm{m}}$ | 0.033 s$^{-1}$ | Obtained from fit |
| mRNA production | $\beta$ | 0.1 s$^{-1}$ | Obtained from fit |
| Translation rate | $\alpha$ | 0.03–0.2 s$^{-1}$ | Obtained from fit |
| Dimerization | $k_{\mathrm{p}}$ | 1.38s$^{-1}$ | *Stamatakis and Zygourakis, 2011* |
| Monomerization | $k_{\mathrm{m}}$ | 0.000002s$^{-1}$ | *Stamatakis and Zygourakis, 2011* |

## Data analysis

Data analysis is performed using a modified version of the Matlab code Schnitzcells (*Rosenfeld et al., 2005*). We use this code to segment the phase images of each sample to identify single cells. Mean pixel intensities of YFP and mCherry signals are extracted from the segmented phase mask for each individual cell using regionprops, an inbuilt function in matlab. The background fluorescence is calculated by averaging the mean intensity of the inverse phase mask upon eroding the regions around the segmented cell masks. The background fluorescence value of a particular frame was subtracted from the mean pixel intensity of cells in the same frame (see Appendix 1). Finally, the autofluorescence value were calculated using the same procedure for cells that do not express either YFP or mCherry and the average autofluorescence value of these cells is subtracted from each measured YFP or mCherry value. Resulting mean pixel intensity of mCherry signal was corrected for the crosstalk from YFP signal. Crosstalk between different channels can be measured by determining the difference between the autofluorescence of a strain without a given fluorophore in the presence of the other fluorophore (highly expressed). We find that under our microscope 0.25% ($\gamma_{\mathrm{cross}} = 0.0025$) of YFP signals can be seen in the mCherry channel whereas mCherry channel has no crosstalk in the YFP channel. Hence, we correct for this crosstalk by subtracting the mean pixel intensity of YFP signal times the $\gamma_{\mathrm{cross}}$ from the mean pixel intensity of mCherry signal. The per-pixel fluorescence values of mCherry and YFP of each cell is then multiplied by the area of the cell to account for the total fluorescence. Fold-change in expression of the mCherry and YFP is calculated by dividing the corresponding values of the constitutive strains (discussed in Appendix 4). At least 500 individual cells were analyzed per sample and binned according to the mCherry values. Any bin with less than 50 data points is excluded. Unless otherwise stated, each data point represents the bootstrapped mean of all data points in a given bin and the error bar represents the standard deviation of the bootstrapped mean.

## Acknowledgements

We wish to thank Rob Phillips, Griffin Chure, Manuel Razo-Mejia, Amir Mitchell, Job Dekker, Marian Walhout, and Michael Lee for helpful discussions. We thank Dr. Jeffrey Bailey for providing us with qubit for DNA quantification. We thank Kenan Murphy for his valuable suggestions on protein degradation tags. We declare no conflict of interest. Funding: Research reported in this publication was supported by NIGMS of the National Institutes of Health under award R35GM128797. Author contributions:  MA did the computational analysis; VP performed all the experiments; SC provided the necessary supervision for the computational setup; RB conceptualized the experiments and drafted the manuscript. All data and codes are backed up in UMASS server. We have uploaded the full source code to repeat the simulations in the following link: GitHub link.

## Additional information

### Funding

| Funder | Grant reference number | Author |
| --- | --- | --- |
| National Institute of General Medical Sciences | R35GM128797 | Md Zulfikar Ali Vinuselvi Parisutham Robert C Brewster |

The funders had no role in study design, data collection and interpretation, or the decision to submit the work for publication.

### Author contributions

Md Zulfikar Ali, Conceptualization, Software, Formal analysis, Investigation, Visualization, Methodology, Writing - original draft, Writing - review and editing; Vinuselvi Parisutham, Conceptualization, Resources, Formal analysis, Validation, Investigation, Visualization, Writing - original draft, Writing - review and editing; Sandeep Choubey, Conceptualization, Software, Investigation, Methodology, Writing - original draft, Writing - review and editing; Robert C Brewster, Conceptualization, Data

curation, Supervision, Funding acquisition, Writing - original draft, Project administration, Writing - review and editing

### Author ORCIDs
Md Zulfikar Ali ![ORCID] https://orcid.org/0000-0002-7054-0059
Vinuselvi Parisutham ![ORCID] https://orcid.org/0000-0002-0349-4072
Sandeep Choubey ![ORCID] https://orcid.org/0000-0002-7387-6148
Robert C Brewster ![ORCID] https://orcid.org/0000-0002-7656-4086

### Decision letter and Author response
Decision letter https://doi.org/10.7554/eLife.56517.sa1
Author response https://doi.org/10.7554/eLife.56517.sa2

## Additional files

### Supplementary files
• Source data 1. Simulation data. Data generated through simulations for *Figure 2A-B* , *Figure 3B-D*, *Figure 4B-F* and *Figure 5A-C*.

• Transparent reporting form

### Data availability
Microscopy data was deposited to the Image Data Resource under accession number idr0095. Code used to generate figures and simulation code is available on github at https://github.com/zulfikgp/Autoregulation (copy archived at https://github.com/elifesciences-publications/Autoregulation).

The following dataset was generated:

| Author(s) | Year | Dataset title | Dataset URL | Database and Identifier |
|---|---|---|---|---|
| Ali MZ, Parisutham V, Choubey S, Brewster RC | 2020 | Microscopy data for Inherent regulatory asymmetry emanating from network architecture in a prevalent autoregulatory motif | https://idr.openmicroscopy.org/search/?query=Name:95 | Image Data Resource, idr0095 |

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

## Appendix 1

### Sensitivity in choosing the background values

The local background of each image is subtracted from individual cells of that image, rather than using a global average over every position. Getting a precise quantitative measurement of fluorescence values is important especially for the tagged strains as their mCherry signal can be only several counts above autofluorescence. The background fluorescence can be influenced by factors such as the local thickness of the agarose pad and positional effects due to the glass dish (which can have small local defects). As shown in *Appendix 1—figure 1*, a no fluorescent strain corrected using the local fluorescence (calculated by making an inverse mask of each frame, excluding regions with cell, and calculating the mean intensity of the background) of each frame produces a tight, symmetric distribution of cell fluorescence with the mean centered near 0 when compared to using the mean value of no fluorescent strain. In other words, many of the YFP or mCherry signals that appear high in the autofluorescence samples also have higher than average backgrounds and thus accounting for this image to image difference is important. Hence, for all experiments we have used the local background fluorescence of each frame to correct for the autofluorescence of cells in the corresponding frame and excluding frames with too high variation in the background fluorescence.

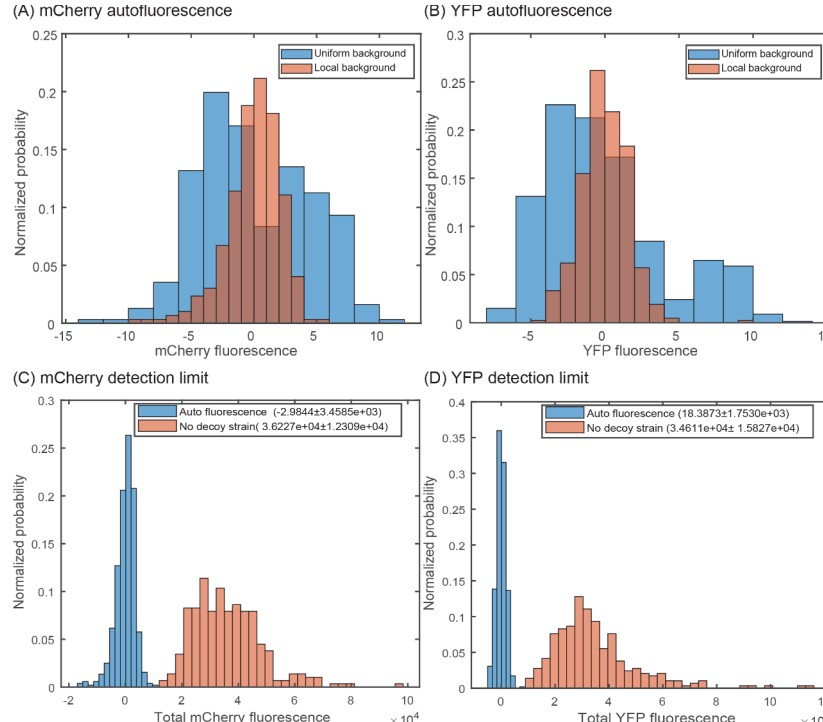

**Appendix 1—figure 1.** Accounting for local variation in background fluorescence. Histogram of single-cell autofluorescence levels of (**A**) mCherry or (**B**) YFP fluorescence in a strain without the YFP and mCherry casettes. The blue bars are calculated as the fluorescence level subtracted from the average across the entire sample (nine different fields of view). The red bars are calculated by first removing the local background fluorescence from cells at each position before subtracting the remaining signal from the average. The wide distribution seen in the blue bars is owed largely to local differences in background fluorescence and is removed by accounting for position-to-position variability. (**C,D**) Histogram showing the minimal detection limit (in a no decoy strain) for mCherry (**C**) and YFP (**D**) compared to an autofluorescence strain.

## Appendix 2

## Cell growth rate in different media

Cell growth rate is measured in strain HG105 growing in a 50 mL flask at 37°C and at 250 rpm. Samples are collected at precise time points and OD600 is measured (see *Appendix 2—figure 1C*). Doubling time is calculated by first interpolating the intermediate time points from the measurements of OD600 and with the single exponential robust fit function in Matlab (see *Appendix 2—figure 1A*). *Appendix 2—figure 1B* shows the scaling in cell area (measured in pixel units) in different media in accordance with the previous literature (*Jun et al., 2018*). Interestingly, the strain with 5X decoy plasmid has a strikingly different area (from other strains) in glucose minimal media possibly indicating sickness due to the presence of multiple arrays of Oid binding site. Hence, results of 5X decoy strain is excluded from the data set for glucose minimal media.

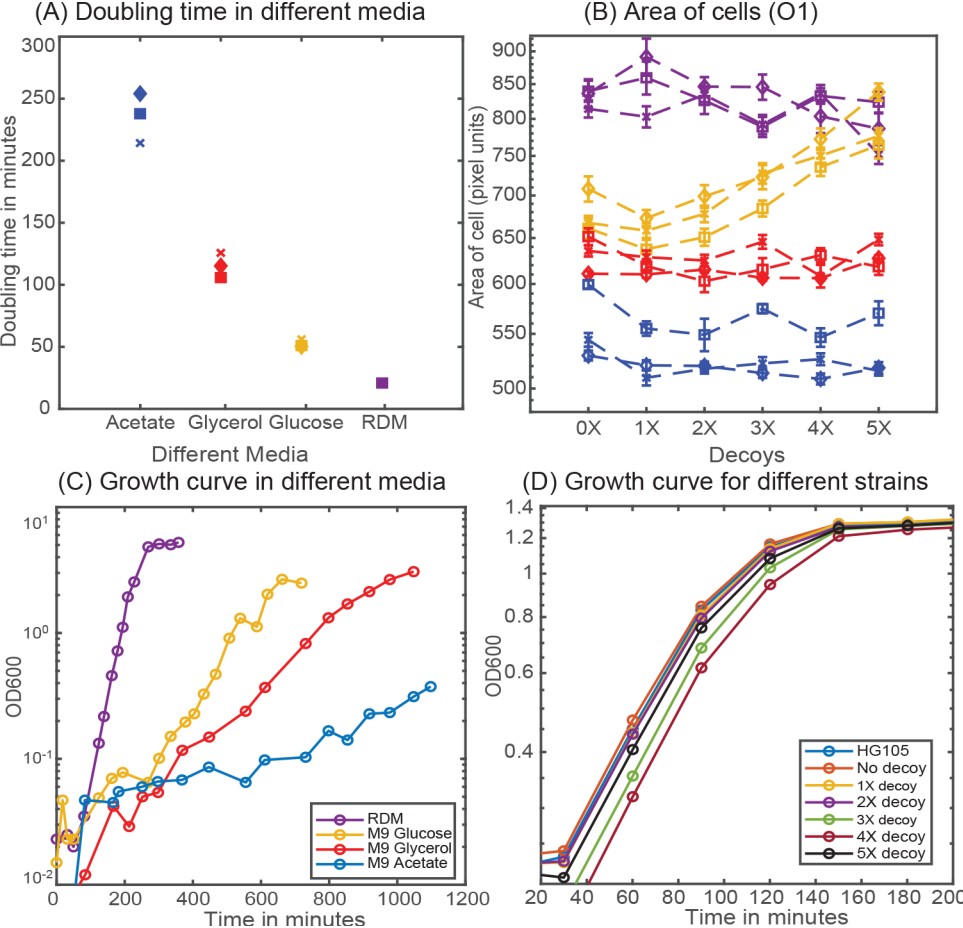

**Appendix 2—figure 1.** Cellular physiology in different media. (**A**) Doubling time of HG105 in different media used in this study. (**B**) Consistent with the literature there is a scaling of cell area in different media in accordance with their growth rate. Strains with 4X and 5X decoys growing in glucose minimal media have a drastically different cell area. (**C**) Plot showing the growth curves for the strain HG105 grown in M9-minimal media with glucose, glycerol and acetate or in rich-defined media. (**D**) Plot showing the growth curves in rich-defined media for strains carrying in different decoy plasmid. Cells are grown in TECAN machine (maintained at 37°C) in a 96-well plate with constant shaking and measurements are made every 30 min.

## Appendix 3

### Quantification of plasmid copy number

Five different variants of Oid decoy arrays (carrying 1, 2, 3, 4 and 5 binding sites for Oid, respectively) are inserted in the intergenic region between the origin of replication and ampicillin cassette of the pZE plasmid. Plasmid copy number is quantified in qPCR measurements using primers that targets a 90 bp-intergenic region in the plasmid backbone immediately upstream of the site of insertion of our decoy array. The total number of decoys can then be estimated by multiplying the measured copy number of pZE plasmid backbone with the number of binding sites in the decoy array. As shown in *Appendix 3—figure 1A*, pZE plasmid backbone had similar copy number in strains with different decoy arrays except for strains carrying the 5X decoy array plasmid. Copy number of 5X-decoy array plasmid is significantly higher when compared to strains carrying other decoy array plasmids. This difference is primarily due to a reduced CFU/mL obtained (see *Appendix 3—figure 1C*) for strains carrying the 5X decoy arrays; the number of molecules of plasmid per reaction is uniform across different strains (see *Appendix 3—figure 1B*). It is not clear if this is due to this sample actually containing less cells or if it is due to a reduced ability to recover and separate these cells (which tend to clump and stick more in microscopy imaging) in the plating assay. This may lead to over-prediction of the copy number of 5X decoy plasmid. Hence, we excluded the 5X-decoy plasmid data in *Figure 2A*. The average (± standard deviation) number of decoy binding arrays in different strains are: 39 ± 8, 96 ± 17, 134 ± 25, 245 ± 40, and 607 ± 47, respectively.

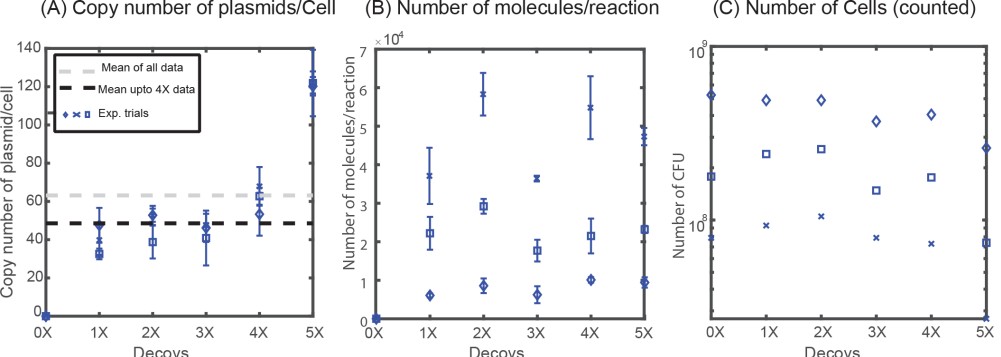

**Appendix 3—figure 1.** Quantification of plasmid copy number. (**A**) Copy number of decoy array plasmids measured in M9-Glucose minimal media. (**B**) Number of molecules obtained per qPCR reaction remains constant across different decoy strains (1X, 2X, 3X, 4X, 5X). (**C**) Number of Colony Forming Units (CFU) per mL used to normalize the number of molecules to account for the copy number of plasmids per cell.

## Appendix 4

### Constitutive values for the autoregulatory gene

To compare expression levels between the TF and the target genes, we wish to compare fold-change as an 'apples-to-apples' comparison of the regulation of each gene. To calculate fold-change we must know the constitutive expression of the gene, that is, how much expression is seen in the absence of regulation by TF. In simulation, this is simple to calculate because we can remove any reactions that include TF binding. Experimentally, calculating constitutive expression for the target gene is also relatively straight-forward; we delete the gene expressing LacI-mCherry and measure the same construct in the absence of TF. However, measuring constitutive expression experimentally for an autoregulating gene was more challenging. There are many possible strategies, but all of them come with some complication. In short, we attempted three different strategies which included: (1) IPTG induction (with or without the addition of decoys), (2) mutated LacI to ablate specific binding, (3) mutated binding site sequences (which has the complication that the site is centered at +11 and thus is both close to the promoter and present on the transcript, see *Appendix 4—figure 1A*). In the end, we identified one mutated site (NoO1V1) which faithfully preserved constitutive expression of the target gene in all media studied. Unfortunately, we were not able to find corresponding mutated sites that reproduced expression of promoters bearing O2 or Oid binding sites. As such, for data using those binding sites on the TF gene we have an unknown scaling factor between the x- and y-axis in the fold-change versus fold-change plots which we determine by fitting the glucose data to our simulations (and then hold constant for all other data sets). In the following sections we discuss techniques we tried.

### Allosteric induction with IPTG to achieve constitutive expression

One way to obtain the constitutive values is to exploit the property of the LacI to become less active when bound to small molecules like IPTG. Previous studies indicate that even with the use of IPTG, expression from a stronger binding site (like Oid) cannot be fully rescued when the repressor copy number is high (*Razo-Mejia et al., 2018*). In our experiments, we observed this phenomenon as well. As shown in *Appendix 4—figure 1C-E*, for most strains expressing the TF, the expression of the target could not be fully rescued with 2.5 mM IPTG and decoys. Further increase in IPTG concentration (to up to 10 mM) did not help in increasing the target expression. Hence, allosteric induction with IPTG could not serve as a right constitutive value for our system.

### Use of LacI with mutated DNA recognition domains

We constructed a mutant protein by deleting 10 amino acids (from amino acid 60 to amino acid 70) in the DNA binding domain of LacI. This mutant helped to completely restore the target expression. However, the mCherry level of strains with the mutated LacI-mCherry were significantly lower than the mCherry level of strains with the functional LacI-mCherry. Since, we would expect the expression of the non-functional TF to be higher than the functional, we reason that this did not provide an accurate estimate of the constitutive mCherry level in the LacI-mCherry strain. This discrepancy may originate from many possible sources such as a change to the stability of the mRNA/protein or a possible alteration to the spectral property of mCherry (which is directly fused to LacI). In the end, we were unable to find a suitable LacI mutant without this feature.

### Use of binding sequence insensitive to LacI

*Oehler et al., 1994* has reported inactivated O1 site (NoO1V1) that has close consensus to O1 binding sequence but does not allow LacI binding. We verified that the expression of YFP from the promoter with NoO1V1 is comparable to the expression of YFP from O1 regulated promoter (in the absence of any LacI) but is lower than the expression from O2 and Oid regulated promoters (*Appendix 4—figure 1B*). Although expression alone does not guarantee that all intermediate steps are precisely the same, we believe this construct gives accurate measurements of constitutive expression for the TF and target genes. We used TF and target with NoO1V1 binding sequence as our constitutive strain to normalize expression from any O1 regulated genes in our experiments. We also tried other forms of mutations on the NoO1V1-binding site (*Appendix 4—figure 1A*) in order to obtain

mutants that relieves *lacI* repression and restore expression of Oid or O2 sequence but with no success.

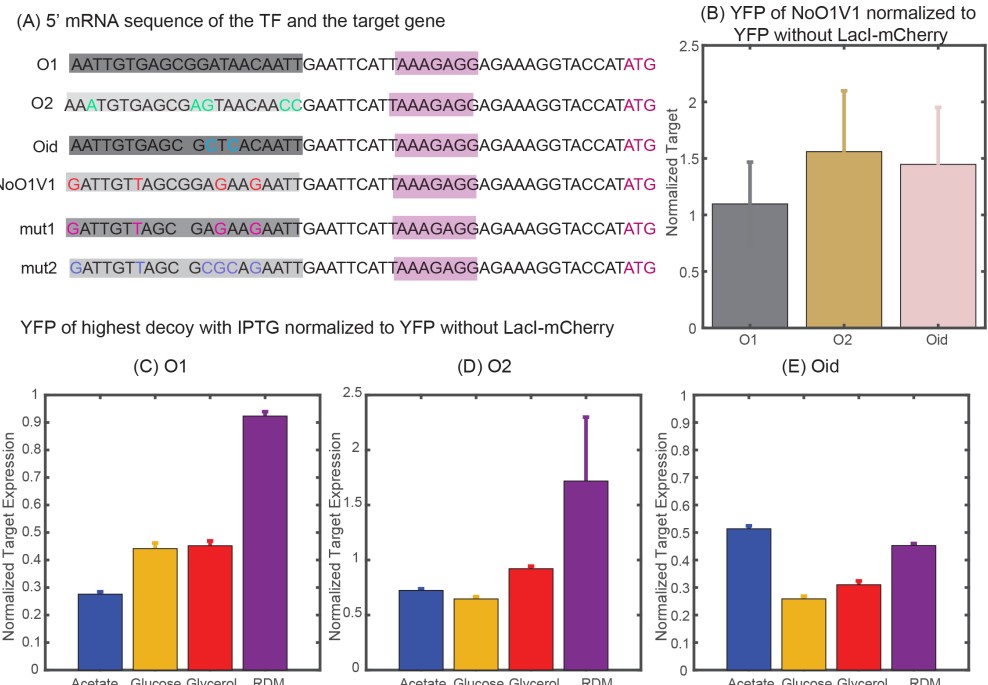

**Appendix 4—figure 1.** Determining constitutive expression of YFP and mCherry. (**A**) 5′ mRNA sequence of the TF and the target genes. The binding site for the TF is carried in the mRNA sequence and is highlighted in shaded dark grey boxes with base changes for different binding sites coded in multicolor. mut1 and mut2 are the two variant binding sites that are designed with mutations similar to NoO1V1 but with Oid site length. However, such changes do not achieve constitutive unregulated expression similar to O2 or Oid. (**B**) Plot showing YFP expressed from NoO1V1 regulated promoter normalized to YFP expressed from promoter regulated with O1, O2 or Oid. (**C-E**) Plot showing the effect of 2.5 mM IPTG in relieving YFP expression form O1 (**C**), O2 (**D**) or Oid. (**E**) Regulated promoter and with 5X decoy plasmids. As indicated in the plot IPTG is not sufficient to restore complete expression of YFP in different media and hence cannot be used as a measure of constitutive expression.

## Appendix 5

### Copy number difference and diffusion limitation of TF

Copy number variation of genes along the long axis of the chromosome and the diffusion limitation of LacI-mCherry could be suggested as a significant contributor to the asymmetry between TF and the target. *E. coli* can initiate multiple replication events (depending on the division rate in the given media) and hence different genes along the chromosome will experience a different copy number in a given time. For instance, *E. coli* growing in RDM (with a division rate of 22 min) will have a copy number of 4 at the *ybcN* locus (where the TF gene is integrated) and a copy number of 3.6 at the *galK* locus (where the target gene is integrated) as described by *Cooper and Helmstetter, 1968*. We believe that the use of fold-change as the measurement of expression helps to reduce the influence of copy number effects (since both the regulated and unregulated measurements have the same copy number). However, the effects may not be linear and LacI has been shown to suffer from diffusion limitation from its origin of synthesis (*Kuhlman and Cox, 2012*). Hence, we tested our system by placing the TF and the target genes integrated next to each other at the *gspI* locus. As evident from *Appendix 5—figure 1*, there is no significant contribution of the copy number difference between TF and target or diffusion limitation of TF on the phenomenon of asymmetry observed in our negatively-autoregulated SIM motif.

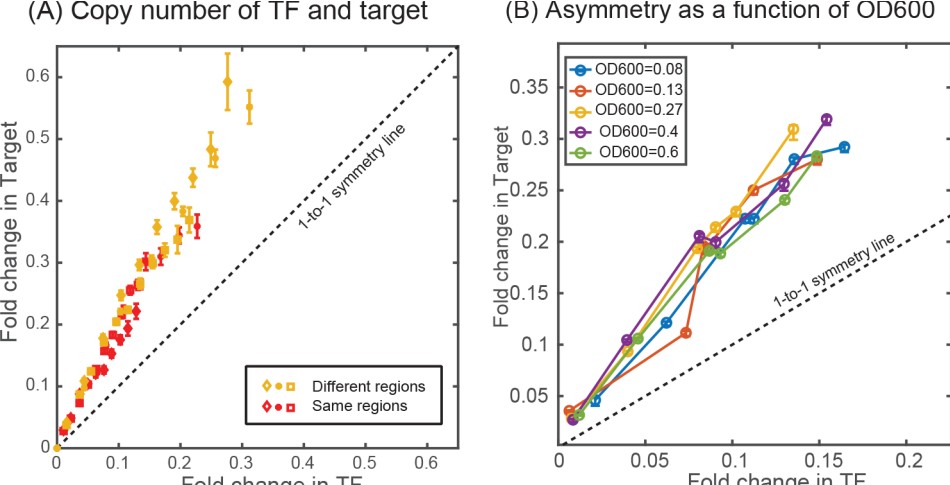

**Appendix 5—figure 1.** Effect of copy number difference on asymmetry. Comparison of asymmetry in strain where the TF and the target genes are located either at two different regions of the chromosome (*ybcN* for TF and *galK* for target, shown in yellow data points) or when it is present together in the chromosome (at the *gspI* locus, shown in red data points). (**B**) Plot showing the measurement of asymmetry in glucose-minimal media at different optical density (OD600) in the exponential phase.

## Appendix 6

### Deterministic solution

Using the assumptions of equilibrium mass-action kinetics, the deterministic counterpart of the negative autoregulation system described in the main text and *Appendix 6—figure 2A* can be written as

$$
\begin{aligned}
\frac{dX}{dt} &= \alpha m_x - \gamma X + 2k_m R - 2k_p X^2, \\
\frac{dR}{dt} &= -k_m R + k_p X^2 - \gamma R - k_{on} R P_{fx} - k_{on} R P_{fy} - k_{on} R N_f + k_{off,x}(1 - P_{fx}) \\
&\quad + k_{off,y}(1 - P_{fy}) + k_{off,d}(N - N_f), \\
\frac{dY}{dt} &= \alpha m_y - \gamma Y, \\
\frac{dP_{fx}}{dt} &= -k_{on} R P_{fx} + (k_{off,x} + \gamma)(1 - P_{fx}), \\
\frac{dP_{fy}}{dt} &= -k_{on} R P_{fy} + (k_{off,y} + \gamma)(1 - P_{fy}), \\
\frac{dN_f}{dt} &= -k_{on} R N_f + (k_{off,d} + \gamma)(N - N_f), \\
\frac{dm_x}{dt} &= \beta P_{fx} - \gamma_m m_x, \\
\frac{dm_y}{dt} &= \beta P_{fy} - \gamma_m m_y.
\end{aligned}
\tag{A6-1}
$$

Here, $X$ is the concentration of free TF monomer, $Y$ is the concentration of target protein, and $R$ is the concentration of TF dimer. $m_x, m_y, P_{fx}(P_{ox}), P_{fy}(P_{oy}), N$, and $N_f(N_o)$ are TF mRNA, target mRNA, free (bound) TF-promoter, free (bound) target-promoter, total concentration of decoy sites, and concentration of free (bound) decoy sites, respectively. Inherent in the equations are the assumptions of the conservation for the concentration of binding sites , that is $P_{fx} + P_{ox} = 1$, $P_{fy} + P_{oy} = 1$, and $N_f + N_o = N$. The right hand side of the equations can be set to zero to obtain the steady state values for all the components.

$$
\begin{aligned}
P_{fx} &= \frac{k_{off,x} + \gamma}{k_{on}R + k_{off,x} + \gamma} = \frac{1}{1 + \sigma_1 R}, \\
P_{fy} &= \frac{k_{off,y} + \gamma}{k_{on}R + k_{off,y} + \gamma} = \frac{1}{1 + \sigma_2 R}, \\
N_f &= \frac{N(k_{off,d} + \gamma)}{k_{on}R + k_{off,d} + \gamma} = \frac{N}{1 + \sigma_3 R}, \\
m_x &= \frac{\beta}{\gamma_m} P_{fx}, \\
m_y &= \frac{\beta}{\gamma_m} P_{fy}, \\
0 &= \alpha m_x - \gamma X + 2k_m R - 2k_p X^2, \\
0 &= -k_m R + k_p X^2 - \gamma(R + P_{ox} + P_{oy} + N_o) \\
Y &= \frac{\alpha\beta}{\gamma\gamma_m} P_{fy} = \frac{\alpha\beta}{\gamma\gamma_m} \frac{1}{1 + \sigma_2 R},
\end{aligned}
\tag{A6-2}
$$

where $\sigma_i = k_{on}/(k_{off,i} + \gamma)$. The concentration of total TF protein can be expressed as a sum of free TF monomer, TF dimer bound to each promoter, and TF dimers bound to the decoys sites

$$
\begin{aligned}
X_{Total} &= X + 2(R + P_{ox} + P_{oy} + N_o), \\
&= \frac{\alpha}{\gamma} m_x, \\
&= \frac{\alpha\beta}{\gamma\gamma_m} \frac{1}{1 + \sigma_1 R}.
\end{aligned}
\tag{A6-3}
$$

The fold-change of the TF and target expression, thus can be obtained by dividing $X_{Total}$ and $Y$ with the constitutive expression, that is, $\alpha\beta/\gamma\gamma_m$ which yields,

$$\text{FC}_{\text{TF}} = \frac{1}{1 + \sigma_1 R} = \frac{1}{1 + \frac{k_{\text{on}}}{k_{\text{off},x} + \gamma} R}, \tag{A6-4}$$

$$\text{FC}_{\text{Target}} = \frac{1}{1 + \sigma_2 R} = \frac{1}{1 + \frac{k_{\text{on}}}{k_{\text{off},y} + \gamma} R} \tag{A6-5}$$

It is worth noting that both TF and target protein follows $1/(1 + R^*)$, where $R^* = R k_{\text{on}}/(k_{\text{off}} + \gamma)$ is the reduced free TF concentration, which is equivalent to the thermodynamic solution (*Weinert et al., 2014*). When the unbinding rates of TF and target are identical, each of them follow the same fold-change curve irrespective of the competition from other decoy sites. In *Appendix 6—figure 1C*, we plot the fold-change for TF and target with $k_{\text{off},x}$ corresponding to O1 binding site and $k_{\text{off},y}$ corresponding to O1 (yellow), O2 (purple), and Oid (blue). It can be seen from the figure that when the off-rates are identical the fold-change curve follows one-to-one line showing no asymmetry which is in contrast with the results obtained using stochastic simulations and experimental results. Furthermore, both the transient and steady state behavior of mean fold-change of TF and target obtained from deterministic solution deviate from the stochastic behavior (see *Appendix 6—figure 1B*). Importantly, when autoregulation is removed from the simulation, the deterministic and stochastic solutions agree precisely (*Appendix 6—figure 1B* inset).

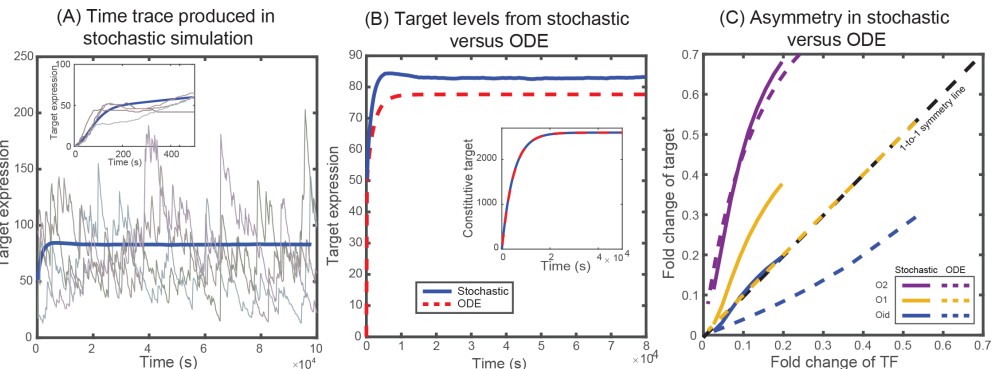

**Appendix 6—figure 1.** Solutions from stochastic simulation and from deterministic ODEs. (**A**) Representative time traces of target expression in individual cells (grey shades) from stochastic simulations. Blue solid line represents the mean behavior averaged over $5 \times 10^4$ iterations. Inset shows the transient behavior. (**B**) Plot showing the average target expression in the negative SIM motif from stochastic simulations (solid line) and from solving deterministic ODEs (dashed line). Inset shows that when regulation is removed the average levels are identical for stochastic and deterministic models. (**C**) Plot showing the asymmetry between TF and target expression from using either stochastic simulation (solid lines) or solving deterministic ODEs (dashed lines). The TF is always regulated by O1 binding site, whereas the target is regulated by O1 (yellow), O2 (purple) or Oid (blue) binding sites. The black dashed line represents line of no asymmetry.

## (A) Full model

| Schematic reaction | Stochastic chemical reaction | Deterministic rate |
|---|---|---|
| TF/target gene → TF/target gene + TF/target mRNA | $m_{\text{TF(Target)}} \xrightarrow{\beta} m_{\text{TF(Target)}} + 1$ | $\beta P_{\text{fx(fy)}}$ |
| TF mRNA → TF mRNA + TF monomer | $m_{\text{TF}} \xrightarrow{\alpha} m_{\text{TF}} + X$ | $\alpha m_{\text{x}}$ |
| target mRNA → target mRNA + target protein | $m_{\text{Target}} \xrightarrow{\alpha} m_{\text{Target}} + Y$ | $\alpha m_{\text{y}}$ |
| monomer + monomer → TF dimer | $X + X \xrightarrow{k_{\text{p}}} Z$ | $k_{\text{p}} X^2$ |
| TF dimer → monomer + monomer | $Z \xrightarrow{k_{\text{m}}} X + X$ | $k_{\text{m}} Z$ |
| gene + dimer → repressed gene | $Z + P_{\text{TF(Target)}} \xrightarrow{k_{\text{on}}} ZP_{\text{TF(Target)}}$ | $k_{\text{on}} ZP_{\text{fx(fy)}}$ |
| repressed gene → gene + dimer | $ZP_{\text{TF(Target)}} \xrightarrow{k_{\text{off,TF(Target)}}} Z + P_{\text{TF(Target)}}$ | $k_{\text{off,x(off, y)}}(1 - P_{\text{fx(fy)}})$ |
| decoy site + dimer → bound decoy | $Z + N \xrightarrow{k_{\text{on}}} ZN$ | $k_{\text{on}} ZN_f$ |
| bound decoy → decoy site + dimer | $ZN \xrightarrow{k_{\text{off,Decoy}}} Z + N$ | $k_{\text{off,d}}(N - N_{\text{f}})$ |
| mRNA → decayed mRNA | $m_{\text{TF(Target)}} \xrightarrow{\gamma_{\text{m}}} m_{\text{TF(Target)}}$ | $\gamma_m m_{\text{x(y)}}$ |
| monomer or dimer or protein → decayed protein | $X, Y, Z \xrightarrow{\gamma} \phi$ | $\gamma X$  or  $\gamma Y$  or  $\gamma Z$ |
| repressed gene or bound decoy → gene or decoy site | $ZP_{\text{TF(Target)}} \xrightarrow{\gamma} P_{\text{TF(Target)}}$   or   $ZN \xrightarrow{\gamma} N$ | $\gamma(1 - P_{\text{fx(fy)}})$ or $\gamma(N - N_{\text{f}})$ |

## (B) Minimal model to demonstrate asymmetry

| Schematic reaction | Stochastic chemical reaction | Deterministic rate |
|---|---|---|
| TF/target gene → TF/target gene + TF/target prot. | $P_{\text{TF(Target)}} \xrightarrow{\alpha} P_{\text{TF(Target)}} + X(Y)$ | $\alpha P_{\text{fx(fy)}}$ |
| gene + TF → repressed gene | $X + P_{\text{TF(Target)}} \xrightarrow{k_{\text{on}}} XP_{\text{TF(Target)}}$ | $k_{\text{on}} XP_{\text{fx(fy)}}$ |
| repressed gene → gene + TF | $XP_{\text{TF(Target)}} \xrightarrow{k_{\text{off}}} X + P_{\text{TF(Target)}}$ | $k_{\text{off}}(1 - P_{\text{fx(fy)}})$ |
| TF or protein → decayed protein | $X, Y \xrightarrow{\gamma} \phi$ | $\gamma X$  or  $\gamma Y$ |
| repressed gene → gene | $XP_{\text{TF(Target)}} \xrightarrow{\gamma} P_{\text{TF(Target)}}$ | $\gamma(1 - P_{\text{fx(fy)}})$ |

**Appendix 6—figure 2.** List of reactions used in the (**A**) stochastic model and (**B**) in the minimal model.

## Appendix 7

### Maximum asymmetry

The asymmetry in regulation (defined as $\mathrm{FC}_{\mathrm{TF}} - \mathrm{FC}_{\mathrm{Target}}$) is a function of all the rates describing the system and number of decoy binding sites. For a given set of rates ($k_{\mathrm{on}}$, $k_{\mathrm{off}}$, $\gamma$, $\gamma_m$) as the decoy number is varied the asymmetry first increases, attains a maximum and then approaches zero for infinite number of decoy binding sites (see **Appendix 7—figure 1**). The maximum asymmetry for a given set of rates is this peak asymmetry observed as decoy number is varied. In the manuscript, we show a heatmap (**Figure 3E**) to emphasize how this maximum asymmetry depends on the two crucial rate parameters, off-rate of the binding sites ($k_{\mathrm{off}}$ or equivalently binding affinity, since in our model $k_{\mathrm{on}}$ is kept constant) and the degradation of TF molecules ($\gamma$).

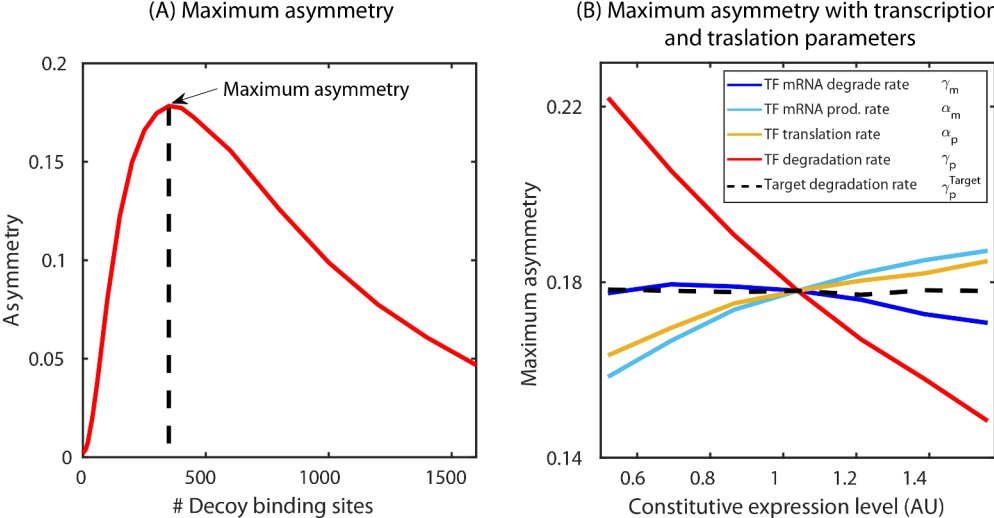

**Appendix 7—figure 1.** Determination of maximum asymmetry. (**A**) Maximum asymmetry in simulation is computed by plotting the asymmetry, difference in fold-change between target and TF, versus number of decoy binding sites in SIM motif. The peak of this asymmetry corresponds to the maximum asymmetry. (**B**) Exploring the model parameters of the TF (mRNA production and degradation; protein production and degradation) that could influence the asymmetry between the TF and the target. Tuning the protein degradation rate (red line) has the maximum influence on the asymmetry between the TF and its target gene.

## Appendix 8

### A minimal model of an autoregulatory gene and a single target gene

The full model in Appendix 6 contains many reactions that are included to more faithfully mirror the biological system we are modeling. However, not all these reactions are necessary to observe the phenomenon of asymmetry which we describe in this manuscript. In this section, we present a reduced model of the extended model of transcription described in Materials and methods to show that the asymmetry in TF and target expression stems from the network architecture and not due to the intermediate steps of transcription and the presence of excess decoy binding sites. We consider an autoregulatory gene whose protein product $X$ inhibits its own expression and also represses a single target gene with protein product $Y$. To reduce the complexity, the protein is made directly from the gene with no intermediates (eliminating translation rates and mRNA decay rates). In this system, the TF, $X$, acts as a monomer and binds to its own gene with rate $k_{\text{on}}$ and unbinds with rate $k_{\text{off}}$. Similarly, the TF ($X$) binds and unbinds from the target gene with the same rates. Both the TF gene and target gene in free state (not bound with TF) produces their protein with rate $\alpha$ which degrades with rate $\gamma$ (dilution through cell division). The reactions describing this reduced model are listed in *Appendix 6—figure 2B*. We implement the simulations using stochastic simulation algorithms as described in Materials and methods section.

Next, we write a set of deterministic coupled ODEs corresponding to the reactions described above which is given by

$$
\begin{aligned}
\frac{dX}{dt} &= \alpha P_{\text{fx}} - \gamma X - k_{\text{on}}XP_{\text{fx}} - k_{\text{on}}XP_{\text{fy}} + k_{\text{off}}(1 - P_{\text{fx}}) + k_{\text{off}}(1 - P_{\text{fy}}), \\
\frac{dY}{dt} &= \alpha P_{\text{fy}} - \gamma Y, \\
\frac{dP_{\text{fx}}}{dt} &= -k_{\text{on}}XP_{\text{fx}} + (k_{\text{off}} + \gamma)(1 - P_{\text{fx}}), \\
\frac{dP_{\text{fy}}}{dt} &= -k_{\text{on}}XP_{\text{fy}} + (k_{\text{off}} + \gamma)(1 - P_{\text{fy}}).
\end{aligned}
\tag{A8-1}
$$

Here, $X$ is the concentration of free TF and $Y$ is the concentration of target protein. $P_{\text{fx}}(P_{\text{ox}})$ and $P_{\text{fy}}(P_{\text{oy}})$ are free (bound) TF-promoter, free (bound) target-promoter, respectively. Inherent in the equations are the assumptions of the conservation for the concentration of binding sites, that is $P_{\text{fx}} + P_{\text{ox}} = 1$, $P_{\text{fy}} + P_{\text{oy}} = 1$. To obtain the steady state values of TF and target expression the right hand side of the equations is set to zero which yield

$$
\begin{aligned}
P_{\text{fx}} &= \frac{k_{\text{off}} + \gamma}{k_{\text{on}}X + k_{\text{off}} + \gamma} = \frac{1}{1 + \sigma X}, \\
P_{\text{fy}} &= \frac{k_{\text{off}} + \gamma}{k_{\text{on}}X + k_{\text{off}} + \gamma} = \frac{1}{1 + \sigma X}, \\
X &= \frac{\alpha}{\gamma}P_{\text{fx}} - P_{\text{ox}} - P_{\text{oy}}, \\
Y &= \frac{\alpha}{\gamma}P_{\text{fy}} = \frac{\alpha}{\gamma}\frac{1}{1 + \sigma X},
\end{aligned}
\tag{A8-2}
$$

where $\sigma = k_{\text{on}}/(k_{\text{off}} + \gamma)$. Total TF concentration, $X_{\text{Total}}$, can be expressed as the sum of free TF and TFs bound to each promoter

$$
\begin{aligned}
X_{\text{Total}} &= X + P_{\text{ox}} + P_{\text{oy}} \\
&= \frac{\alpha}{\gamma}P_{\text{fx}} \\
&= \frac{\alpha}{\gamma}\frac{1}{1 + \sigma X}.
\end{aligned}
\tag{A8-3}
$$

The fold-change of the TF and target expression, thus can be obtained by dividing $X_{\text{Total}}$ and $Y$ by the constitutive expression, that is, without any regulation, $C_0 = \alpha/\gamma$ which yields,

$$
\text{FC}_{\text{TF}} = \frac{1}{1 + \sigma X} = \frac{1}{1 + \frac{k_{\text{on}}}{k_{\text{off}} + \gamma}X},
\tag{A8-4}
$$

$$\mathrm{FC_{Target}} = \frac{1}{1+\sigma X} = \frac{1}{1+\frac{k_{\mathrm{on}}}{k_{\mathrm{off}}+\gamma}X}. \tag{A8-5}$$

As was shown previously in section Appendix 6, both TF and target protein follows $1/(1+\sigma X)$ and show no asymmetry in regulation.

Furthermore, solving *Equation A8-2* we get the free TF expression as,

$$X = \frac{-1-2\sigma+\sqrt{(1+2\sigma)^2+4C_0\sigma}}{2\sigma}. \tag{A8-6}$$

## Appendix 9

### Chemical master equation (CME) for the minimal model

The chemical master equation governing the dynamics of the expression for TF and target gene for the minimal model discussed in Appendix 8 (also shown in *Appendix 6—figure 2B*) is given by

$$
\begin{aligned}
\frac{dP_{00}(n,m,t)}{dt} &= \alpha[P_{00}(n-1,m,t) - P_{00}(n,m,t) + P_{00}(n,m-1,t) - P_{00}(n,m,t)] \\
&\quad + \gamma[(n+1)P_{00}(n+1,m,t) - nP_{00}(n,m,t) + (m+1)P_{00}(n,m+1,t) \\
&\quad - mP_{00}(n,m,t) + P_{01}(n,m,t) + P_{10}(n,m,t)] + k_{\text{off}}[P_{01}(n-1,m,t) \\
&\quad + P_{10}(n-1,m,t)] - 2k_{\text{on}}nP_{00}(n,m,t), \\
\frac{dP_{01}(n,m,t)}{dt} &= \alpha[P_{01}(n-1,m,t) - P_{01}(n,m,t)] + \gamma[(n+1)P_{01}(n+1,m,t) \\
&\quad - nP_{01}(n,m,t) + (m+1)P_{01}(n,m+1,t) - mP_{01}(n,m,t) \\
&\quad + P_{11}(n,m,t) - P_{01}(n,m,t)] + k_{\text{off}}[P_{11}(n-1,m,t) - P_{01}(n,m,t)] \\
&\quad + k_{\text{on}}[(n+1)P_{00}(n+1,m,t) - nP_{01}(n,m,t)], \\
\frac{dP_{10}(n,m,t)}{dt} &= \alpha[P_{10}(n,m-1,t) - P_{10}(n,m,t)] + \gamma[(n+1)P_{10}(n+1,m,t) \\
&\quad - nP_{10}(n,m,t) + (m+1)P_{10}(n,m+1,t) - mP_{10}(n,m,t) \\
&\quad + P_{11}(n,m,t) - P_{10}(n,m,t)] + k_{\text{off}}[P_{11}(n-1,m,t) - P_{10}(n,m,t)] \\
&\quad + k_{\text{on}}[(n+1)P_{00}(n+1,m,t) - nP_{10}(n,m,t)], \\
\frac{dP_{11}(n,m,t)}{dt} &= \gamma[(n+1)P_{11}(n+1,m,t) - nP_{11}(n,m,t) + (m+1)P_{11}(n,m+1,t) \\
&\quad - mP_{11}(n,m,t) - 2P_{11}(n,m,t)] - 2k_{\text{off}}P_{11}(n,m,t) + \\
&\quad k_{\text{on}}[(n+1)P_{01}(n+1,m,t) + (n+1)P_{10}(n+1,m,t)]
\end{aligned}
$$

(A9-1)

Here, $P_{ij}(n,m,t)$ is the probability of having $n$ TF protein and $m$ target protein at any instant of time $t$ in the state $(i,j)$. $i$ and $j$ denotes the occupancy of the TF promoter and target promoter, respectively. A value of 0 indicates that the promoter of TF/target gene is occupied by a TF. A value of 1, similarly indicates a promoter which is free to express.

Summing *Equation A9-1* over all values of $(m,n)$ we get the rate equation for occupancy defined as $S_{ij} = \sum_{n,m=0}^{\infty} P_{ij}$ in each state

$$
\begin{aligned}
\frac{dS_{00}}{dt} &= (\gamma + k_{\text{off}})(S_{01} + S_{10}) - 2k_{\text{on}}\langle n \rangle_{00}, \\
\frac{dS_{01}}{dt} &= (\gamma + k_{\text{off}})(S_{11} - S_{01}) + k_{\text{on}}[\langle n \rangle_{00} - \langle n \rangle_{01}], \\
\frac{dS_{10}}{dt} &= (\gamma + k_{\text{off}})(S_{11} - S_{10}) + k_{\text{on}}[\langle n \rangle_{00} - \langle n \rangle_{10}], \\
\frac{dS_{11}}{dt} &= -2(\gamma + k_{\text{off}})S_{11} + k_{\text{on}}[\langle n \rangle_{01} + \langle n \rangle_{10}].
\end{aligned}
$$

(A9-2)

Multiplying both sides of *Equation A9-1* by $n$ and summing over all values of $(m,n)$ we get the time evolution of free TF protein in each state ($\langle n \rangle_{ij} = \sum_{m,n} nP_{i,j}(n,m,t)$)

$$
\begin{aligned}
\frac{d\langle n \rangle_{00}}{dt} &= \alpha S_{00} - \gamma\langle n \rangle_{00} + \gamma[\langle n \rangle_{01} + \langle n \rangle_{10}] + k_{\text{off}}[\langle n+1 \rangle_{01} + \langle n+1 \rangle_{10}] - 2k_{\text{on}}\langle n^2 \rangle_{00} \\
\frac{d\langle n \rangle_{01}}{dt} &= \alpha S_{01} - \gamma\langle n \rangle_{01} + \gamma[\langle n \rangle_{11} - \langle n \rangle_{01}] + k_{\text{off}}[\langle n+1 \rangle_{11} - \langle n \rangle_{01}] + k_{\text{on}}[\langle n(n-1) \rangle_{00} - \langle n^2 \rangle_{01}] \\
\frac{d\langle n \rangle_{10}}{dt} &= -\gamma\langle n \rangle_{10} + \gamma[\langle n \rangle_{11} - \langle n \rangle_{10}] + k_{\text{off}}[\langle n+1 \rangle_{11} - \langle n \rangle_{10}] + k_{\text{on}}[\langle n(n-1) \rangle_{00} - \langle n^2 \rangle_{10}] \\
\frac{d\langle n \rangle_{11}}{dt} &= -\gamma\langle n \rangle_{11} - 2\gamma\langle n \rangle_{11} - 2k_{\text{off}}\langle n \rangle_{11} + k_{\text{on}}[\langle n(n-1) \rangle_{01} + \langle n(n-1) \rangle_{10}].
\end{aligned}
$$

(A9-3)

Similarly, multiplying both sides of *Equation A9-1* by $m$ and summing over all values of $(m,n)$ we obtain the time evolution of target protein in each state ($\langle m \rangle_{ij} = \sum_{m,n} mP_{i,j}(n,m,t)$)

$$\frac{d\langle m \rangle_{00}}{dt} = \alpha S_{00} - \gamma \langle m \rangle_{00} + \gamma[\langle m \rangle_{01} + \langle m \rangle_{10}] + k_{\text{off}}[\langle m \rangle_{01} + \langle m \rangle_{10}] - 2k_{\text{on}}\langle mn \rangle_{00}$$

$$\frac{d\langle m \rangle_{01}}{dt} = -\gamma \langle m \rangle_{01} + \gamma[\langle m \rangle_{11} - \langle m \rangle_{01}] + k_{\text{off}}[\langle m \rangle_{11} - \langle m \rangle_{01}] + k_{\text{on}}[\langle mn \rangle_{00} - \langle mn \rangle_{01}]$$

$$\frac{d\langle m \rangle_{10}}{dt} = \alpha S_{10} - \gamma \langle m \rangle_{01} + \gamma[\langle m \rangle_{11} - \langle m \rangle_{10}] + k_{\text{off}}[\langle m \rangle_{11} - \langle m \rangle_{10}] + k_{\text{on}}[\langle mn \rangle_{00} - \langle mn \rangle_{01}]$$

$$\frac{d\langle m \rangle_{11}}{dt} = -\gamma \langle m \rangle_{11} - 2\gamma \langle m \rangle_{11} - 2k_{\text{off}}\langle m \rangle_{11} + k_{\text{on}}[\langle mn \rangle_{01} + \langle mn \rangle_{10}].$$

(A9-4)

The rate equation for total number of TF (sum of the free TFs in each state and the bound TFs in state 2, 3, and 4) and the total target protein can be written as

$$\frac{d\langle n \rangle}{dt} = \frac{d}{dt}[\langle n \rangle_{00} + \langle n \rangle_{01} + \langle n \rangle_{10} + \langle n \rangle_{11} + S_{01} + S_{10} + 2S_{11}]$$
$$= \alpha(S_{00} + S_{01}) - \gamma \langle n \rangle$$
$$\frac{d\langle m \rangle}{dt} = \frac{d}{dt}[\langle m \rangle_{00} + \langle m \rangle_{01} + \langle m \rangle_{10} + \langle m \rangle_{11}]$$
$$= \alpha(S_{00} + S_{10}) - \gamma \langle m \rangle.$$

(A9-5)

The steady state expression for total TF and target can be obtained by setting *Equation A9-5* to zero which yields

$$\langle n \rangle_{\text{ss}} = \frac{\alpha}{\gamma}(S_{00} + S_{01}) = C_0(S_{00} + S_{01}),$$
$$\langle m \rangle_{\text{ss}} = \frac{\alpha}{\gamma}(S_{00} + S_{10}) = C_0(S_{00} + S_{10}),$$

(A9-6)

where $C_0 = \alpha/\gamma$ is the constitutive protein expression. The asymmetry defined as the difference of fold change in expression of target and TF gene expression is given by

$$\text{Asymmetry} = \text{FC}_{\text{Target}} - \text{FC}_{\text{TF}},$$
$$= \frac{\langle m \rangle_{\text{ss}}}{C_0} - \frac{\langle n \rangle_{\text{ss}}}{C_0},$$
$$= S_{10} - S_{01}.$$

(A9-7)

The asymmetry in TF and target regulation simply depends on the difference of occupancy in the states where TF gene is bound and where the target gene is bound. Furthermore, the steady state occupancies are given by (setting *Equations A9-2* to zero)

$$S_{01} = S_{11} + \frac{k_{\text{on}}}{(\gamma + k_{\text{off}})}[\langle n \rangle_{00} - \langle n \rangle_{01}],$$
$$S_{10} = S_{11} + \frac{k_{\text{on}}}{(\gamma + k_{\text{off}})}[\langle n \rangle_{00} - \langle n \rangle_{10}],$$
$$S_{11} = \frac{k_{\text{on}}}{2(\gamma + k_{\text{off}})}[\langle n \rangle_{01} + \langle n \rangle_{10}].$$

(A9-8)

The asymmetry using *Equations A9-7 and A9-8* is then given by

$$\text{Asymmetry} = \frac{k_{\text{on}}}{(\gamma + k_{\text{off}})}[\langle n \rangle_{01} - \langle n \rangle_{10}].$$

(A9-9)

*Equation A9-9* clearly demonstrates that the asymmetry in TF and target expression arises from the difference in the free TF concentration in state 2 (only target gene bound) and state 3 (only TF gene bound). Analytical expression for free TFs in different state cannot be determined explicitly as it can be seen from the *Equations A9-3 and A9-4* that the mean protein ($\langle n \rangle, \langle m \rangle$) depends on the higher order moments ($\langle n^2 \rangle, \langle mn \rangle$) which then depends on the next higher order moments and so on.

## Appendix 10

### Modified ODEs for the minimal model

The asymmetry, as explained in the main text and evident from *Equation A9-9*, appears due to the difference in the TF concentration when only the TF gene is occupied and when only the target gene is occupied. The general deterministic approach does not capture this asymmetry due to the mean field assumption of uniform TF concentration in all the states. To incorporate the difference in TF concentration in the deterministic model we now specifically assume the four state model; (1) both the TF gene and target gene are free to express, (2) TF gene is bound by TF, (3) target gene is bound by TF, and (4) both the genes are bound by TF. The number of cells in each state are $S_1, S_2, S_3$, and $S_4$ and the total population ($S$) is constant. The free TF and total target protein number in each states are $(n_1, m_1)$, $(n_2, m_2)$, $(n_3, m_3)$, and $(n_4, m_4)$ such that the average free TFs in each cell is $\langle n \rangle_i = n_i / S_i$ and average target protein in each cell is $\langle m \rangle_i = m_i / S_i$. State 1 switches to state 2 and 3 when a free TF binds to the free promoter of TF gene or target gene. State 2 and state 3 switch to state 1 when a bound TF unbinds or degrade from the gene. State 2 and state 3 also switch to state 4 due to TF binding. Finally, state 4 switches to state 2 and state 3 when a bound TF unbinds or degrade from the gene.

Change in cell number due to the reactions that switch the cells from state *i* to state *j* causing an increase(state *j*) or decrease (state *i*) in the cell population per unit time are

$$\text{Binding}: \quad k_{\text{on}} \langle n_i \rangle S_i = k_{\text{on}} \frac{n_i}{S_i} S_i = k_{\text{on}} n_i$$
$$\text{Unbinding}: \quad k_{\text{off}} S_i \qquad \qquad \qquad \text{(A10-1)}$$
$$\text{Degradation of TF from gene}: \quad \gamma S_i$$

When a TF binds to a promoter of TF gene or target gene in state *i* switching the cells to state *j* the number of free TF of the cells in state *j* increases by the $(\langle n \rangle_i - 1)$ times the number of cell switched ($k_{\text{on}} n_i$) and the number of target protein increases by $\langle m \rangle_i k_{\text{on}} n_i$. It is to be noted that a binding event decreases the average free TF pool by one in the cells which switch from state *i* to state *j*. In the process, the cells in state *i* loses $\langle n \rangle_i k_{\text{on}} n_i$ number of free TFs and $\langle m \rangle_i k_{\text{on}} n_i$ number of target. Similarly, when a TF unbinds from a promoter switching state *i* to state *j* the number of free TFs of cells in state *j* increases by $(\langle n \rangle_i + 1)$ times the number of cell switched ($k_{\text{off}} S_i$) and the number of free TFs of each cell in state *i* goes down by $\langle n \rangle_i$ times the number of cell switched. The target protein number of cells in state *i* goes down by $\langle m \rangle_i k_{\text{off}} n_i$ and increase by the same amount in state *j*. Degradation of bound TF changes the free TF number by $\langle n \rangle_i \gamma S_i$ and target protein number by $\langle m \rangle_i \gamma S_i$. The change in protein number for all the reactions are listed in *Appendix 10—table 1*.

The set of ODEs describing the time evolution of the cell populations ($S_i$) in each state is then given by

$$\frac{dS_1}{dt} = -2k_{\text{on}} n_1 + (k_{\text{off}} + \gamma)(S_2 + S_3),$$
$$\frac{dS_2}{dt} = k_{\text{on}} n_1 - k_{\text{on}} n_2 + (k_{\text{off}} + \gamma)(S_4 - S_2),$$
$$\frac{dS_3}{dt} = k_{\text{on}} n_1 - k_{\text{on}} n_3 + (k_{\text{off}} + \gamma)(S_4 - S_3), \qquad \text{(A10-2)}$$
$$\frac{dS_4}{dt} = k_{\text{on}} n_2 + k_{\text{on}} n_3 - 2(k_{\text{off}} + \gamma)S_4.$$

The rate equations for free TF number can be written as

$$\frac{dn_1}{dt} = \alpha S_1 - \gamma n_1 + k_{off}(n_2 + S_2) + k_{off}(n_3 + S_3) - 2k_{on}\frac{n_1^2}{S_1} + \gamma(n_2 + n_3),$$

$$\frac{dn_2}{dt} = \alpha S_2 - \gamma n_2 + k_{off}(n_4 + S_4) - k_{off}n_2 + k_{on}\frac{n_1(n_1 - S_1)}{S_1} - k_{rmon}\frac{n_2^2}{S_2} + \gamma(n_4 - n_2),$$

$$\frac{dn_3}{dt} = -\gamma n_3 + k_{off}(n_4 + S_4) - k_{off}n_3 + k_{on}\frac{n_1(n_1 - S_1)}{S_1} - k_{on}\frac{n_3^2}{S_3} + \gamma(n_4 - n_3),$$

$$\frac{dn_4}{dt} = -\gamma n_4 - 2k_{off}n_4 + k_{on}\frac{n_2(n_2 - S_2)}{S_2} + k_{on}\frac{n_3(n_3 - S_3)}{S_3} - 2\gamma n_4,$$

(A10-3)

and the rate equations for target protein number is given by

$$\frac{dm_1}{dt} = \alpha S_1 - \gamma m_1 + k_{off}m_2 + k_{off}m_3 - 2k_{on}\frac{m_1 n_1}{S_1} + \gamma(m_2 + m_3),$$

$$\frac{dm_2}{dt} = -\gamma m_2 + k_{off}m_4 - k_{off}m_2 + k_{on}\frac{m_1 n_1}{S_1} - k_{on}\frac{m_2 n_2}{S_2} + \gamma(m_4 - m_2),$$

$$\frac{dm_3}{dt} = \alpha S_3 - \gamma m_3 + k_{off}m_4 - k_{off}m_3 + k_{on}\frac{m_1 n_1}{S_1} - k_{on}\frac{m_3 n_3}{S_3} + \gamma(m_4 - m_3),$$

$$\frac{dm_4}{dt} = -\gamma m_4 - 2k_{off}m_4 + k_{on}\frac{m_2 n_2}{S_2} + k_{on}\frac{m_3 n_3}{S_3} - 2\gamma m_4.$$

(A10-4)

Using *Equations A10-2–A10-4*, the rate equations for total TF and target protein can be written as

$$\frac{dn}{dt} = \frac{d}{dt}(n_1 + n_2 + n_3 + n_4 + S_2 + S_3 + 2S_4),$$
$$= \alpha(S_1 + S_2) - \gamma n,$$
$$\frac{dm}{dt} = \frac{d}{dt}(m_1 + m_2 + m_3 + m_4),$$
$$= \alpha(S_1 + S_3) - \gamma m.$$

(A10-5)

The steady state concentration for total TF and target protein is obtained by setting *Equation A10-5* to zero which gives

$$n_{ss} = \frac{\alpha}{\gamma}(S_{1,ss} + S_{2,ss}) = C_0(S_{1,ss} + S_{2,ss}),$$

$$m_{ss} = \frac{\alpha}{\gamma}(S_{1,ss} + S_{3,ss}) = C_0(S_{1,ss} + S_{3,ss}).$$

(A10-6)

Here, $C_0 = \alpha/\gamma$ is the protein number of unregulated gene (constitutive expression). The steady state number of cells in states in terms of free TF number can be obtained by setting *Equation A10-2* to zero and is given by

$$S_{1,ss} = S - (S_{2,ss} + S_{3,ss} + S_{4,ss}) = S - \frac{k_{on}}{2(k_{off} + \gamma)}(4n_{1,ss} + n_{2,ss} + n_{3,ss}),$$

$$S_{2,ss} = \frac{k_{on}}{2(k_{off} + \gamma)}(2n_{1,ss} - n_{2,ss} + n_{3,ss})$$

$$S_{3,ss} = \frac{k_{on}}{2(k_{off} + \gamma)}(2n_{1,ss} + n_{2,ss} - n_{3,ss})$$

$$S_{4,ss} = \frac{k_{on}}{2(k_{off} + \gamma)}(n_{2,ss} + n_{3,ss}).$$

(A10-7)

Setting $S = 1$ converts the number of cells ($S_i$) to occupancy of the cell in each state and $n_i$, $m_i$ to fractional average of free TF and target protein per cell, that is, $n_i = n_{ss}S_i$ and $m_i = m_{ss}S_i$. The asymmetry defined as the difference of fold change in expression of target and TF gene expression is given by

$$
\begin{aligned}
\text{Asymmetry} &= \text{FC}_{\text{Target}} - \text{FC}_{\text{TF}}, \\
&= \frac{m_{\text{ss}}}{C_0} - \frac{n_{\text{ss}}}{C_0}, \\
&= S_{3,\text{ss}} - S_{2,\text{ss}}, \\
&= \frac{k_{\text{on}}}{k_{\text{off}} + \gamma}(n_{2,\text{ss}} - n_{3,\text{ss}}).
\end{aligned}
\tag{A10-8}
$$

It is important to note that the same set of ODEs (*Equations A10-2–A10-4*) can be derived from CME by setting the variance and covariance of protein number in each state to zero. This modified ODEs predicts asymmetry between the TF and target expressions as shown in *Appendix 10—figure 1A*, however, the predicted asymmetry doesn't match quantitatively with the CME predictions (see *Appendix 10—figure 1B*). This discrepancy arises because of the fluctuations in the protein number in each state which is not considered in the modified ODEs.

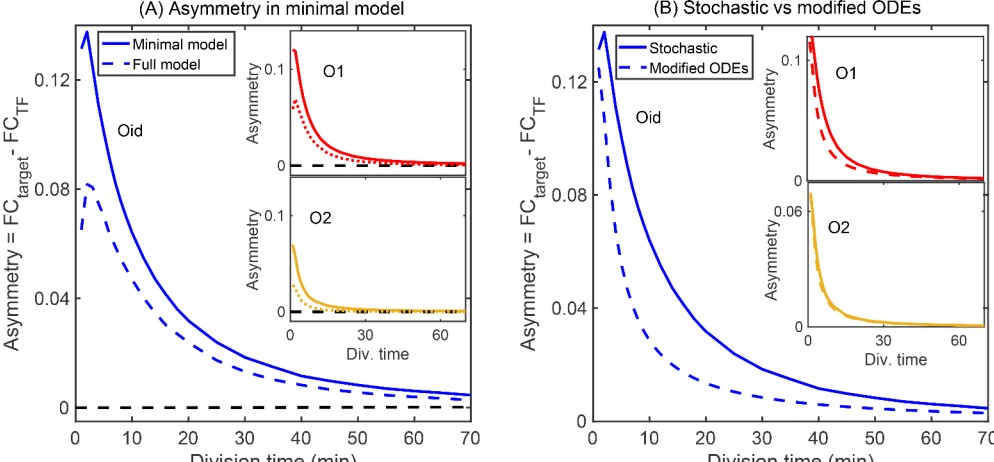

**Appendix 10—figure 1.** Minimal model of autoregulation. (**A**) Asymmetry predicted from a minimal model without intermediate transcription steps and decoy binding sites using stochastic simulations (solid lines in blue, red and yellow for Oid, O1 and O2 binding sites, respectively). The asymmetry follows similar trend as predicted in the complete stochastic model (shown as dashed lines). Stronger binding site (Oid, shown in solid blue line) shows higher asymmetry than a weak binding site (O2, shown in solid yellow line). Also, asymmetry decreases as the growth rate is increased. Black dashed line corresponds to the deterministic counterpart of the stochastic reaction systems. Again, we do not find any asymmetry in TF and target regulation from the deterministic solution. (**B**) Modified ODEs with the inclusion of four states each having a different TF concentration predict asymmetry (dashed lines for different binding sites; Oid [blue], O1 [red], and O2 [yellow]). However, the quantitative values disagrees from the stochastic simulations of minimal model shown as solid lines.

**Appendix 10—table 1.** Change in free TF and target protein number for the reactions describing the minimal model.

| Reaction | Increase in free TF | Decrease in free TF |
|---|---|---|
| Production in active state | $\alpha S_i$ | - |
| Degradation of free TF | - | $\gamma n_i$ |
| Binding | $(\frac{n_i}{S_i} - 1)k_{\text{on}}n_i = k_{\text{on}}\frac{n_i(n_i - S_i)}{S_i}$ | $\frac{n_i}{S_i}k_{\text{on}}n_i = k_{\text{on}}\frac{n_i^2}{S_i}$ |
| Unbinding | $(\frac{n_i}{S_i} + 1)k_{\text{off}}S_i = k_{\text{off}}(n_i + S_i)$ | $(\frac{n_i}{S_i} + 1)k_{\text{off}}S_i = k_{\text{off}}(n_i + S_i)$ |
| Degradation of TF from gene | $\frac{n_i}{S_i}\gamma S_i = \gamma n_i$ | $\frac{n_i}{S_i}\gamma S_i = \gamma n_i$ |
| | Increase in target | Decrease in target |

*Continued on next page*

*Appendix 10—table 1 continued*

| Reaction | Increase in free TF | Decrease in free TF |
|---|---|---|
| Production in active state | $\alpha S_i$ | - |
| Degradation of target | - | $\gamma m_i$ |
| Binding | $\frac{m_i}{S_i} k_{\text{on}} n_i = k_{\text{on}} \frac{m_i n_i}{S_i}$ | $\frac{m_i}{S_i} k_{\text{on}} n_i = k_{\text{on}} \frac{m_i n_i}{S_i}$ |
| Unbinding | $\frac{m_i}{S_i} k_{\text{off}} S_i = k_{\text{off}} m_i$ | $\frac{m_i}{S_i} k_{\text{off}} S_i = k_{\text{off}} m_i$ |
| Degradation of TF from gene | $\frac{m_i}{S_i} \gamma S_i = \gamma m_i$ | $\frac{m_i}{S_i} \gamma S_i = \gamma m_i$ |

## Appendix 11

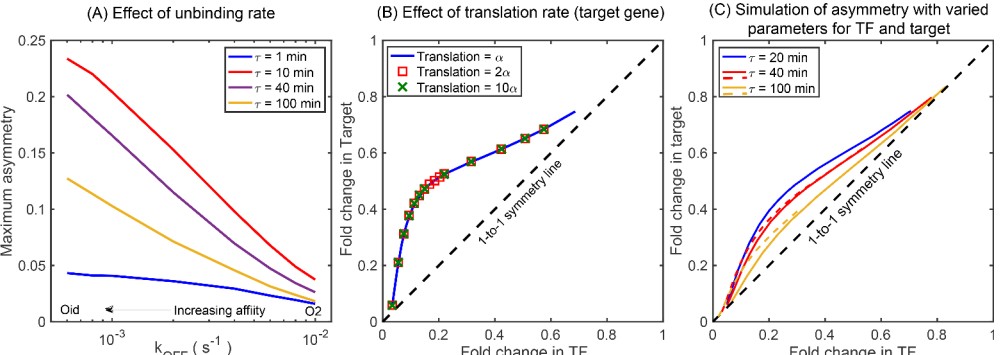

**Appendix 11—figure 1.** Simulations showing the effects of rate parameters on asymmetry. (**A**) Effect of TF unbinding rate ($k_{OFF}$) on asymmetry. Irrespective of the $k_{OFF}$, the maximum asymmetry decreases monotonically. (**B**) Asymmetry is not affected by difference in translation rate between the TF gene and the target gene. Blue solid curve represents asymmetry obtained from simulations where the translation rate of TF gene and the target gene is exactly same. The data points are generated with a translation rate of target gene twice (red square) and ten times (green cross) that of the TF gene and fall exactly on the blue curve showing no deviation. (**C**) Asymmetry for different growth rate (τ) with varying transcription rate, translation rate, and mRNA stability. Stochastic simulation performed using the kinetic parameters listed in *Bremer and Dennis, 2008* for τ being 20 (blue line), 40 (red line), and 100 (yellow line) minutes. Dashed lines show the asymmetry for τ = 40 min and 100 min for the rate parameters same as τ = 20 min. The qualitative ordering and features of the asymmetry curve is not impacted by the changes in the kinetic parameters such as transcription rate, translation rate, and mRNA stability due to change in growth rates.

## Appendix 12

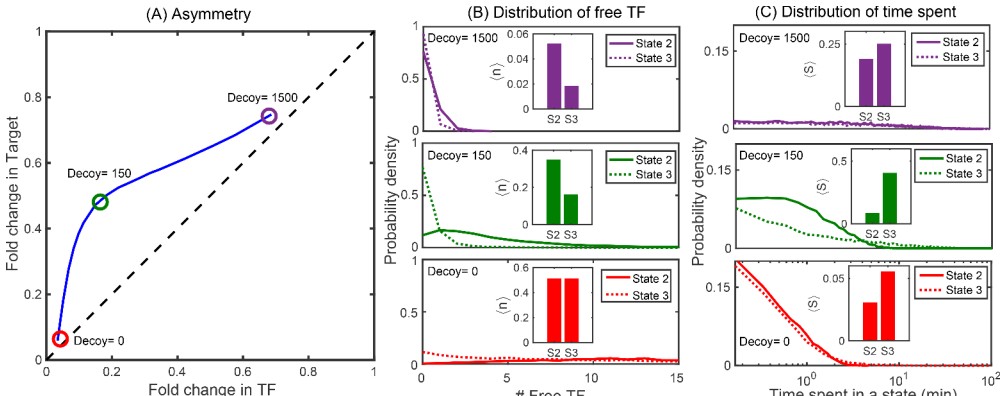

**Appendix 12—figure 1.** Distributions of free TFs and time spent in different promoter states. (**A**) Typical asymmetry plot obtained from simulations for Oid binding site with division time τ = 25 min. (**B-C**) Distribution of free TFs and time spent in state 2 (S2) and state 3 (S3) for varying level of asymmetry corresponding to different decoy number as shown in panel (A). The plots in red, green and purple correspond to no decoys (low asymmetry), 150 decoy (maximum asymmetry) and 1500 decoy (low asymmetry). Insets in (B) are steady state fractional average of free TFs in state 2 and state 3 obtained from stochastic simulations using equation $\langle n \rangle = \sum_{m,n} n P_{i,j}$, where $P_{i,j}$ is the probability of having m target protein and n free TF in the promoter state (i,j); see Appendix 9. Insets in (C) are TF-occupancy in state 2 and state 3 defined as $\langle n \rangle = \sum_{m,n} P_{i,j}$.

