## [Decision Letter]

**Acceptance summary:**

Combining synthetic biology, live-cell imaging and stochastic mathematical models, this study demonstrates and explains the non-intuitive observation that, in a single-input module, the expression of genes controlled by a self-regulated transcription factor may differ even when both genes have the same binding sites.

**Decision letter after peer review:**

Thank you for submitting your article "Inherent regulatory asymmetry emanating from network architecture in a prevalent autoregulatory motif" for consideration by *eLife*. Your article has been reviewed by two peer reviewers, and the evaluation has been overseen by a Reviewing Editor and Naama Barkai as the Senior Editor. The reviewers have opted to remain anonymous.

The reviewers have discussed the reviews with one another and the Reviewing Editor has drafted this decision to help you prepare a revised submission.

Summary:

The reviewers found the manuscript to be very interesting, in particular the non-intuitive observation that the expression of genes controlled by a self-regulated transcription factor differ even when both genes have the same binding sites. No additional experiments are needed, but both reviewers have asked that some issues be clarified before the paper can be accepted for publication. Please see the reviewer comments below and answer them point-by-point in a revised manuscript.

Reviewer #1:

Using a combined computational and experimental approach, the authors rigorously evaluate a single-input module that shows gene regulation asymmetry for genes with identical regulatory sequences. The authors find that the effect can only be reproduced in a stochastic setting and features such as network size and dilution rate are identified as determinants of the degree of asymmetry. This paper is both well written and executed and its implications in understanding synthetic as well as natural regulatory systems make this well suited for a broad audience. If the authors address some concerns, I would recommend this manuscript for publication in *eLife*.

1) The experimental methods were not completely clear. Overnight cells were diluted into fresh media and incubated to early log before sampling. Microscopy was then used to quantitate the fluorescence of cells on agarose pads. Are cells assumed to have reached steady-state before sampling and if not could this affect the results? The addition of any time course data that the authors may already have to show that steady-state was achieved would strengthen the data.

2) Why were the results for the 5x decoy plasmid included in the manuscript? The low CFU/mL observed in Appendix 3 and the increased cell size when grown in glucose medium (Appendix 2) suggests that it may have a deleterious effect on the cells, in which case perhaps it would be better to leave out the 5x results. Also, since the estimated copy number for this condition seems to be an outlier as the authors state that the 5x data was not used for copy number estimation in Appendix 3, how was the decoy number for this condition estimated and what number is actually used? The authors should also address whether the different decoy plasmids cause differences in growth rate if known.

3) Another possible explanation for asymmetry could be a difference in translation rates. Even though the promoter and 5'UTRs are the same, regions at the start of the coding sequence (YFP vs LacI) could interact differently with the 5'UTRs. The authors used YFP to show that the promoter with the NoO1v1 operator has similar expression levels to that of the regulated promoter in the absence of LacI. In this case, normalization would not rule out such an effect. The authors could use simulation to comment on how much of the asymmetry could be attributed to differences in translation rates.

4) The authors show that asymmetry exhibits a non-monotonic dependence on TF half-life (degradation rate). The authors should comment on the possible origin of this non-monotonicity. My intuition is that degradation rate allows the TF to reach steady-state faster. Therefore, faster degradation would allow its concentration to better "track" the regulatory state of the TF gene leading to larger asymmetry. But why would an increase in degradation rate above a certain point lead to less asymmetry?

5) The degradation rate experiments required the use of a *sspB* deletion strain which affects the cell globally. Did the authors observe any changes in growth rate due to the mutation?

6) In the Introduction, the authors claim their aim is to unravel the influence of network size and connectivity on gene regulation. However, their results don't necessarily generalize to more complex networks and different network motifs. It would strengthen the paper if the authors could comment on this in the Discussion.

7) Both in the Introduction and in Figure 1 (panels B and C), much emphasis is put on the role of SIMs in coordination of gene expression timing. The authors should comment on how their results are expected to affect such functions or whether there are any other potential consequences of the discovered asymmetry for natural systems (maybe related to some of the examples given in the Introduction).

Reviewer #2:

In this work, the authors use synthetic biology and live-cell imaging to experimentally characterize the most common regulatory motif in *E. coli*, the single-input module (SIM). The authors find a non-intuitive result: the expression level of a self-regulatory transcription factor (TF) and their target genes differ, even when both genes have the same binding sites. Using a combination of theory and simulation, they formulate a stochastic model that can account for their experimental observations. A simpler, deterministic model is unable to recapitulate their results.

The paper has sound logic, with a straightforward narrative that allows the reader to grasp the author's model and relate the experimental results with the simulations and theory. The appendices answered several of the questions that arose while reading the manuscript (for example, the effect of genome position and the local diffusion limitation of transcription factors). The results shown are an elegant example of the power of theory and modeling to make sense of non-intuitive biological phenomena. I would be glad to see this work published in *eLife*.

– The authors calculate the apparent or effective fold change, given that the minimum fluorescent unit measurable does not correspond to zero TFs but the detection limit. The fact that the experiments do not have single-molecule sensitivity should not make a big difference if the detection limit is small. Do the authors have some experimental evidence that this is the case? If no experiments are available, do they have a theoretical expectation of the error they could incur due to such effects? Would the effect still be negligible if both fluorescent proteins have different detection limits? Would we expect different results if they would have used yfp for the TF and mCherry for the target gene?

– Figure 4B seems critical for understanding their model. In principle, showing a single realization of their stochastic model with the transitions between microstates would be useful to convey the idea "at the single-cell level." However, given that the difference between the residency times of states 2 and 3 is small, such a plot may not be clear. Can the authors show, in addition to Figure 4B (maybe in an appendix), a complementary figure with the distributions of "Number of free TFs" and "Time in state"? If the authors have other ideas on how to emphasize this insight, I suggest adding them to the manuscript.

– In the caption of Figure 3, (B-C), the authors write: "the actual free TF measured in simulation." Using the word "measured" to refer to simulation results is confusing. Using "obtained" or "calculated" or omit the word, as used in the titles of Figures 3B-C, would be preferable.

– Figure 5D looks very crowded, which makes it hard to compare distributions between samples and figure panels. Could the authors simplify the plot, for example, by removing the bar plots and leaving the lines (or vice-versa)? If the authors keep the bars, they may want to check whether removing the black outlines and applying transparency makes samples easier to distinguish.

– The authors use "single-input" and "single input" interchangeably. It would be better to be consistent across the text.

---

## [Author Response]

Reviewer #1:[…]1) The experimental methods were not completely clear. Overnight cells were diluted into fresh media and incubated to early log before sampling. Microscopy was then used to quantitate the fluorescence of cells on agarose pads. Are cells assumed to have reached steady-state before sampling and if not could this affect the results? The addition of any time course data that the authors may already have to show that steady-state was achieved would strengthen the data.

A detailed experimental method for growing cells is described in Materials and methods. Cells were grown overnight in LB and diluted to an initial OD of ~0.002 into desired media and grown until cells reach an OD of 0.2-0.4 before cells were harvested for microscope. This allows for cells to undergo roughly 8 divisions before measurement, ensuring that memory from the overnight growth in LB is minimized while capturing cells in a reproducible mid-log phase. To demonstrate that cells are in mid-log phase at this point, we now included time course data for the growth of cells in different media in Appendix 2—figure 1C. Furthermore, we have also added in Appendix 5—figure 1B, asymmetry measured over a range of ODs in glucose minimal media. The figure demonstrates that our cells are measured in steady state and that the results are consistent over this range of OD.

2) Why were the results for the 5x decoy plasmid included in the manuscript? The low CFU/mL observed in Appendix 3 and the increased cell size when grown in glucose medium (Appendix 2) suggests that it may have a deleterious effect on the cells, in which case perhaps it would be better to leave out the 5x results. Also, since the estimated copy number for this condition seems to be an outlier as the authors state that the 5x data was not used for copy number estimation in Appendix 3, how was the decoy number for this condition estimated and what number is actually used? The authors should also address whether the different decoy plasmids cause differences in growth rate if known.

We had considered this option previously. In fact, in the original submission we did not include the 5x data in Figure 2A (Fold-change vs decoy number) simply due to this difficulty. This is the only figure in the manuscript where the copy number of the plasmid is required. Though we don’t know the underlying causes of the sickness phenotype in 5x glucose, the growth rate was not significantly different from strains carrying other decoy plasmids and we decided to include them in our analysis. Please refer to Appendix 2—figure 1D of the revised manuscript for some additional discussion of this point. We believe the primary impact of their increased size is our inability to quantify cellular concentrations using OD. However, given the reviewers concern we decided to remove 5x glucose from all combined data figures (i.e. those where data points are pooled and fold-change vs fold-change is plotted such as 3D, 4F, 5A, 5B and 5C). We have left the histogram of Figure 5D since the data in question can be easily “parsed” out. Please note that this change did not significantly impact any of our results.

3) Another possible explanation for asymmetry could be a difference in translation rates. Even though the promoter and 5'UTRs are the same, regions at the start of the coding sequence (YFP vs LacI) could interact differently with the 5'UTRs. The authors used YFP to show that the promoter with the NoO1v1 operator has similar expression levels to that of the regulated promoter in the absence of LacI. In this case, normalization would not rule out such an effect. The authors could use simulation to comment on how much of the asymmetry could be attributed to differences in translation rates.

This is an excellent point and we agree with the reviewer that the translation rate between YFP (target) and LacI-mCherry (TF) can be different. However, since there is no feedback of target protein in controlling the state occupancy or the number of TFs, any change in the translation rate of target, or for that matter any other rates except the binding/unbinding rate, the normalization cancels out the effect. We verified this using simulation and added an additional figure (Appendix 11—figure 1B) to support our claim (red circles and green crosses are simulation with the translation rate of target gene, twice and ten times that of the TF gene). Additionally, this point can be seen from Appendix 7—figure 4B (Maximum asymmetry with transcription and translation rates), where the asymmetry is only affected by changes in the rates associated with the TF gene.

4) The authors show that asymmetry exhibits a non-monotonic dependence on TF half-life (degradation rate). The authors should comment on the possible origin of this non-monotonicity. My intuition is that degradation rate allows the TF to reach steady-state faster. Therefore, faster degradation would allow its concentration to better "track" the regulatory state of the TF gene leading to larger asymmetry. But why would an increase in degradation rate above a certain point lead to less asymmetry?

This is an important point and clearly, we didn’t do a good job in explaining the behavior of asymmetry as a function of the degradation rate. The asymmetry is the difference of occupancy in state 2 (TF gene occupied by TF) and state 3 (target gene occupied by TF). A cell transits out of states 2 or 3 to state 1, either when a TF gets degraded (with rate γ) or unbinds from the promoters of TF or target gene (with rate k_off_). On the other hand, the system transits to state 4 from state 2 or 3 through the binding of a TF molecule to one of the empty promoters. In the small γ limit, the number of TFs in the cells are high, favoring the transition to state 4 very quickly, thereby reducing the residence times of both state 2 and 3. On the other extreme, when γ is large, there is hardly any TF in the first place to enter into state 2 or 3; the cell spends most of the time in state 1. In both the cases, the difference of residence times between state 2 and state 3 is low and hence the asymmetry is small. In the intermediate regime of γ, the number of TFs is optimum to maximize the difference between residence times in state 2 and 3, which leads to maximum asymmetry. We have now added a paragraph in subsection “Dependence of regulatory asymmetry on TF degradation and binding affinity”, see paragraph two.

5) The degradation rate experiments required the use of a sspB deletion strain which affects the cell globally. Did the authors observe any changes in growth rate due to the mutation?

No significant growth difference was observed with the deletion of *sspB* gene. However, it is also important to point out that any experiment using the *sspB* deletion strain uses only data from *sspB* deletion strains including the no fluorescence controls. We have added a sentence stating that deletion of this gene did not account for any significant growth difference.

6) In the Introduction, the authors claim their aim is to unravel the influence of network size and connectivity on gene regulation. However, their results don't necessarily generalize to more complex networks and different network motifs. It would strengthen the paper if the authors could comment on this in the Discussion.

This is a good point. The claim in the Introduction is inadvertently bold, we want to point out that network size can have interesting and unexpected effects on regulation of network motifs. Our results are not generalizable but rather our starting point. We have changed the text: “In this study, we dissect the prevalent gene regulation motif, the single-input module (SIM), to demonstrate the influence of network size and connectivity on the regulation of a network motif.”

We have altered the concluding paragraph of the Discussion: “Finally, here we demonstrate regulatory asymmetry using a specific (but common) regulatory motif. The more general problem of quantifying the role of asymmetry in other network motifs may be an important step in expanding the predictive power of models based on single genes. The broader point that specific genes can be exposed to systematically different levels of regulatory TFs even in the absence of specific cellular mechanisms such as cytoplasmic compartmentalization, protein localization or DNA accessibility is likely more generally relevant. Understanding and quantifying these mechanisms can be an important piece towards improving our ability to predict and design gene regulatory circuits.”

7) Both in the Introduction and in Figure 1 (panels B and C), much emphasis is put on the role of SIMs in coordination of gene expression timing. The authors should comment on how their results are expected to affect such functions or whether there are any other potential consequences of the discovered asymmetry for natural systems (maybe related to some of the examples given in the Introduction).

Currently we discuss this briefly in the Discussion in the sentence quoted below. To further expand upon this point, we have now added the part in red to explicitly highlight the implications:

“This is particularly relevant for the SIM motif because the primary function of the motif, organizing and coordinating gene expression patterns, operates on the premise of differential affinities amongst target genes; here we have shown that the TF gene has an inherent “affinity advantage” due to being exposed to systematically higher TF concentrations than its target genes. This implies that the TF gene will respond “earlier” than expected based on the raw affinity of its binding site and may necessitate weaker sites on autoregulating TF genes in order to achieve similar timing in expression compared to its targets. This may also shed light on the discrepancies in Arg pathway timing between different experiments which have used plasmid reporters (essentially changing network size) or different physiological growth conditions; the asymmetry is critically sensitive to both of these features”.

Reviewer #2:[…]– The authors calculate the apparent or effective fold change, given that the minimum fluorescent unit measurable does not correspond to zero TFs but the detection limit. The fact that the experiments do not have single-molecule sensitivity should not make a big difference if the detection limit is small. Do the authors have some experimental evidence that this is the case? If no experiments are available, do they have a theoretical expectation of the error they could incur due to such effects? Would the effect still be negligible if both fluorescent proteins have different detection limits? Would we expect different results if they would have used yfp for the TF and mCherry for the target gene?

We have added a figure showing single cell fluorescence of the lowest signal (no decoy) for both channels compared to the single cell autofluorescence of each channel. As can be seen, in this case the signal is well separated from the no-fluorescence cells, see Appendix 1—figure 1C and D. We have considered swapping the fluorophores to exclude any possible source of asymmetry due to fluorescent protein properties. However, the LacI-yfp fusion protein we created wasn’t functional. We have tried two different fluorophores for the target (yfp and cfp) but this did not significantly alter our results.

– Figure 4B seems critical for understanding their model. In principle, showing a single realization of their stochastic model with the transitions between microstates would be useful to convey the idea "at the single-cell level." However, given that the difference between the residency times of states 2 and 3 is small, such a plot may not be clear. Can the authors show, in addition to Figure 4B (maybe in an appendix), a complementary figure with the distributions of "Number of free TFs" and "Time in state"? If the authors have other ideas on how to emphasize this insight, I suggest adding them to the manuscript.

We really appreciate the reviewer’s suggestion here. Now, we have added a figure in the Appendix 12 that shows the distributions of the number of free TFs (B) and the times spent (C) in state 2 and 3, corresponding to three different points with varying levels of asymmetry (A). These three points are with zero decoy with low asymmetry (in red), intermediate number of decoys corresponding to the point of maximum asymmetry (in green), and for higher number of decoys (in magenta) where asymmetry again is low. It is evident that when the asymmetry is low the distribution of residence times are similar (red and magenta) but for large asymmetry these distributions become qualitatively distinct (green). It should be noted that the actual residence times can be of the order of seconds to hundreds of minutes depending on the parameters as can be seen in the plot in Appendix 12—figure 1. However, In the manuscript, in Figure 4B, we have reported the occupancy (normalized residency times of each state) and not the actual residence times. The reason for reporting the occupancy is that analytically the difference of occupancy in state 3 and state 2 corresponds to the asymmetry (also mentioned in the manuscript).

– In the caption of Figure 3, (B-C), the authors write: "the actual free TF measured in simulation." Using the word "measured" to refer to simulation results is confusing. Using "obtained" or "calculated" or omit the word, as used in the titles of Figures 3B-C, would be preferable.

We have changed the word “measured” to “obtained” for clarity.

– Figure 5D looks very crowded, which makes it hard to compare distributions between samples and figure panels. Could the authors simplify the plot, for example, by removing the bar plots and leaving the lines (or vice-versa)? If the authors keep the bars, they may want to check whether removing the black outlines and applying transparency makes samples easier to distinguish.

We have added interpolated lines that help visualize the figure in a better way.

– The authors use "single-input" and "single input" interchangeably. It would be better to be consistent across the text.

Thank you for noticing this. We have standardized these terms and now use the term “single-input” consistently.